



# A lab in the field: High-frequency analysis of water quality
# and stable isotopes in streamwater and precipitation
Jana von Freyberg[1,2], Bjørn Studer[1], James W. Kirchner[1, 2]
[1] Department of Environmental Systems Science, ETH Zurich, Zurich, Switzerland
[2] Swiss Federal Research Institute WSL, Birmensdorf, Switzerland
*Correspondence to*: Jana von Freyberg (jana.vonfreyberg@usys.ethz.ch)
**Abstract.** High-frequency measurements of solutes and isotopes ($^{18}$O and $^2$H) in rainfall and streamflow can
shed important light on catchment flow pathways and travel times, but the workload and sample storage artifacts
involved in collecting, transporting, and analyzing thousands of bottled samples severely constrain catchment
studies where conventional sampling methods are employed. However, recent developments towards more
compact and robust analyzers have now made it possible to measure chemistry and water isotopes in the field at
sub-hourly frequencies over extended periods. Here we present laboratory and field tests of a membrane-
vaporization continuous water sampler coupled to a cavity ring-down spectrometer for real-time measurements
of $\delta^{18}$O and $\delta^2$H, combined with a dual-channel ion chromatograph (IC) for synchronous analysis of major
cations and anions. The precision of the isotope analyzer was typically better than 0.03‰ for $\delta^{18}$O and 0.17‰
for $\delta^2$H, for 10min average readings taken at intervals of 30min. Carryover effects were less than 1.2% between
isotopically contrasting water samples for 30min sampling intervals, and instrument drift could be corrected
through periodic analysis of secondary reference standards.
We tested the coupled isotope analyzer / IC system under field conditions by analyzing streamwater and
precipitation every 30min over 28 days in a small catchment. These high-frequency measurements facilitated a
detailed comparison of event-water fractions via end-member mixing analysis with both chemical and isotope
tracers. For two events with relatively dry antecedent moisture conditions, event-water fractions were <20%
based on isotope tracers, but were significantly overestimated (39% to 83%) by the chemical tracers. These
observations, coupled with the storm-to-storm patterns in precipitation isotope inputs and the associated
streamwater isotope response, led to a conceptual hypothesis for runoff generation in the catchment. Under this
hypothesis, the pre-event water that is mobilized by precipitation events may, depending on antecedent moisture
conditions, be significantly shallower, younger, and less mineralized than the deeper, older water that feeds base
flow and thus defines the "pre-event" end-member used in hydrograph separation. This proof-of-concept study
illustrates the potential advantages of capturing isotopic and hydrochemical behavior at high frequency over
extended periods that span multiple hydrologic events.
## 1. Introduction
Environmental tracers are widely used in hydrology to investigate recharge processes, subsurface flow
mechanisms and streamflow components (Leibundgut and Seibert, 2011). The most common environmental
tracers are the naturally occurring stable water isotopes $^{18}$O and $^2$H (Klaus and McDonnell, 2013). Solutes such
as dissolved organic compounds, nutrients, and major ions are also widely used, together with stable isotopes, as
indicators of flowpaths and biogeochemical reactions (e.g., McGlynn and McDonnell, 2003; Vitvar and
Balderer, 1997; Weiler et al., 1999). Environmental tracer studies typically involve manual or automated



sample collection, followed by transport, storage, and subsequent laboratory analysis. The time and effort
involved in sample handling are often a major constraint limiting the frequency and duration of sampling, and
thus the scope of tracer studies.

To date, isotope studies have maintained high sampling frequencies only during a few storm events (e.g.,
Berman et al., 2009; Lyon et al., 2008; Pangle et al., 2013), with the result that only limited ranges of catchment
behavior have been explored. Long-term catchment studies capture a wider range of hydrologic events, but
generally collect water samples at only weekly or monthly intervals for subsequent laboratory analysis (e.g.,
Buso et al., 2000; Darling and Bowes, 2016; Jasechko et al., 2016), making higher-frequency behaviors
unobservable. As pointed out by Kirchner et al. (2004), sampling at intervals much longer much smaller than
the hydrological response times of a catchment may result in significant losses of information. For instance,
sub-daily sampling is required to capture diurnal fluctuations in streamwater hydrochemistry, which reflect in-
stream biological activity (e.g., Hayashi et al., 2012; Rode et al., 2016b). Thus, high-frequency sampling can
help to determine ecological effects or to identify biogeochemical hot spots and hot moments, which are
characterized by disproportionately high reaction rates (e.g., McClain et al., 2003; Vidon et al., 2010). In order
to differentiate hydrological and biogeochemical catchment processes related to different water ages and flow
pathways, long-term monitoring has to be complemented by additional high-frequency hydrochemical and
isotope measurements. So far, only a few long-term studies have sampled streamwater at daily or sub-daily
intervals for on-site measurements or subsequent analysis in the laboratory, such as at Plynlimon, Wales (Neal
et al., 2011), at the Kervidy-Naizin catchment in western France (Aubert et al., 2013) or at the Selke river in
Germany (Rode et al., 2016a). Such studies have yielded fundamental insights into catchment hydrological
behaviour, not only at a wide range of temporal scales but also under varying hydro-climatic conditions (e.g.,
Benettin et al., 2015; Halliday et al., 2013; Harman, 2015; Kirchner and Neal, 2013; Riml and Worman, 2015).

The recent development of compact and robust isotope analyzers has fostered initial attempts to continuously
measure $\delta^{18}O$ and $\delta^2H$ in streamwater or precipitation directly in the field. The only previous field-based
isotope monitoring of 4 contiguous weeks was carried out by Berman et al. (2009) with a customized liquid
water isotope analyzer based on off-axis integrated cavity output spectroscopy (OA-ICOS; Los Gatos Research,
Mountain View, CA, USA), which measured $\delta^{18}O$ and $\delta^2H$ in 90 samples per day. As the system was based on
repeated injections of samples into a vaporizer, daily maintenance (i.e., injection septa change, filter cleaning)
was required to keep it running. An alternative approach uses a semi-permeable membrane to generate water
vapor from a continuous sample throughflow, which is then transferred to a wavelength scanned – Cavity Ring-
Down Spectrometer (CRDS) (e.g., Herbstritt et al., 2012). Munksgaard et al. (2011) developed such a custom-
made diffusion sampler and attached it to a CRDS (Picarro Inc., Santa Clara, CA, USA) to measure $\delta^{18}O$ and
$\delta^2H$ in precipitation at frequencies of up to 30s over a 15day period (Munksgaard et al., 2012).

A similar diffusion sampling system has recently become commercially available (Continuous Water Sampler
Module, or CWS; Picarro Inc., Santa Clara, CA, USA), which allows for quasi-continuous measurements of
$\delta^{18}O$ and $\delta^2H$ in liquid water samples when coupled to a CRDS analyzer. Here we present initial laboratory and
field verification experiments with this device, which we have combined with a dual-channel ion chromatograph





(IC; Metrohm AG, Herisau, Switzerland) for real-time analysis of major cations and anions. Laboratory
experiments quantifying the precision and sample carryover memory effects of this system are presented in
Section 3 below. Section 4 illustrates the practical application of the system in the field using a 28-day
deployment at a small catchment in Switzerland. Section 5 quantifies the fractions of event water that
contributed to the flood hydrograph in eight major precipitation events, illustrating one potential application of
high-frequency isotope tracer measurements.
**2. Methodology**
**2.1 Isotope analysis and ion chromatography**
For the analysis of the stable water isotopes $^{18}$O and $^{2}$H, the Continuous Water Sampler module (CWS; Picarro
Inc., Santa Clara, CA, USA) was coupled to a Wavelength Scanned-Cavity Ring-Down Spectrometer (WS-
CRDS; model L2130-*i*, Picarro Inc., Santa Clara, CA, USA). In the CWS, the water sample flows at a rate of
~1mL min$^{-1}$ through an expanded polytetrafluuoroethylene (ePTFE) membrane tube. This tube is mounted in a
stainless steel chamber that is supplied with dry air to facilitate the steady diffusion of a small fraction of the
through-flowing water as vapor through the membrane. Through the continuous flow of dry air over the outer
surface of the membrane, the vapor is carried directly to the CRDS for isotope analysis. To minimize
temperature-induced fractionation effects, the instrument keeps the temperatures of the membrane chamber and
the inflowing water constant at (± 1 standard deviation) 45±0.1°C and 15±0.1°C, respectively. A solenoid
diaphragm pump situated upstream of the membrane cartridge draws water samples from the sample container
and pushes them through the membrane tube at a flow rate of approximately 1 mL min$^{-1}$. As we show in
Section 3.1 below, preliminary tests showed that this pump is not sufficient for our purposes, so we substituted a
programmable high-precision dosing unit (800 Dosino, Metrohm AG, Herisau, Switzerland) in its place.
Isotopic abundances are reported through the δ notation relative to the VSMOW-SLAP standards. We used the
factory calibration of the isotope analysis system, because only relative isotope values are needed for
quantifying precision, drift, and carryover, and thus the absolute isotope values are unimportant.

Major ions in liquid water samples, i.e. $Na^+$, $K^+$, $NH_4^+$, $Ca^{2+}$, $Mg^{2+}$, $F^-$, $Cl^-$, $NO_3^-$, $SO_4^{2-}$, $PO_4^{3-}$, were analyzed
with an ion chromatograph (IC; model 940 Professional IC Vario, Metrohm AG, Herisau, Switzerland) with a
two-column configuration (Anions: Metrosep A Supp 5 – 250/4.0, Cations: Metrosep c 6 – 250/4.0).
Continuous operation of the instrument was possible due to fully automated eluent generation (941 Eluent
Production Module). To generate the full ion chromatograms of both anions and cations, approximately 28min
were required; thus the sampling interval of the combined analysis system was fixed at 30min.
**2.2 Sample collection and distribution**
The water samples were distributed between the analyzers with high-precision dosing units (800 Dosino, here
called simply 'Dosino'; Metrohm, Herisau, Switzerland). A Dosino contains a programmable piston that fills
and empties a glass cylinder with up to 50 mL of sample at a resolution of 10,000 increments (implying 5 μL
increment$^{-1}$). The design of the dosing unit minimizes the dead volume and thus the potential for sample



carryover. In the base of the glass cylinder sits a rotating valve disc that guides the liquid sample through one of
four ports; thus each Dosino functions as both a switching valve and a syringe pump.

Figure 1 depicts the schematic overview of the automatic sample collection and analysis system, showing how
the different Dosinos distribute precipitation and streamwater samples between the isotope analyzer, the IC and
and an autosampler (which can be programmed to save individual samples for subsequent analysis in the
laboratory). The sampling routine begins with a cleaning step when either the 'P Dosino' or the 'S Dosino'
transports 10 mL of sample water for rinsing to the beaker. The 'Isotope Dosinos' also eject any remaining
sample into the beaker, after which the beaker is emptied. Then, 50 mL of fresh streamwater or precipitation
sample is transported (by either the 'S Dosino' or the 'P Dosino' for streamwater or precipitation, respectively)
into the rinsed beaker, from which one of the 'Isotope Dosinos' draws 30 mL of water and injects it at a flow
rate of 1 mL min$^{-1}$ into the CWS for isotope analysis. The two 'Isotope Dosinos' operate alternatingly to
minimize the time when the sample flow into the CWS is interrupted. Meanwhile, either the 'P Dosino' or the
'S Dosino' takes up another 12 mL of water sample and pumps it through a 0.45 μm tangential filter into the 'IC
Dosino', which discards the first 2 mL of the filtered sample. From the remaining filtered sample, 8 mL are
filled into vials by the autosampler and 2 mL are delivered to the IC for direct ion analysis. During the ion
analysis (ca. 28 min), the 'S Dosino', 'P Dosino' and 'IC Dosino', the autosampler, and all tubing are rinsed
with nanopure water to minimize carryover effects. The entire sampling routine is programmed with the IC
control software MagIC Net (Metrohm, Herisau, Switzerland), which facilitates detailed data logging and
documentation of the sample handling.

### 3. Laboratory experiments


### 3.1 Optimization of sample injection into the Continuous Water Sampler module (CWS)


In the original design of the CWS, water samples are transported by a small solenoid diaphragm pump between
the inlet port and the membrane cartridge at a flow rate of approximately 1 mL min$^{-1}$. During preliminary tests,
however, we observed that raising or lowering the sample container detectably altered the reported isotope
ratios. In order to quantify the sensitivity of the instrument to hydraulic head differences (i.e., the height of the
water table in the sample bottle relative to the waste outlet of the CWS), we changed the elevations of the
sample container relative to instrument while continuously analyzing a single water sample (nanopure water).
We measured vapor concentration, $\delta^{18}O$ and $\delta^{2}H$ for the same water sample at five different elevations, ranging
from 7 cm above to 98 cm below the waste outlet. The end of the waste outlet tube was always freely draining.
Each configuration was measured for one hour and the average values and standard deviations of the
uncalibrated 6s measurements of vapor concentration, $\delta^{18}O$ and $\delta^{2}H$ were calculated from the last 10min of each
1h configuration.

The results of this experiment are summarized in Fig. 2, which shows clear linear relationships between the
hydraulic head differences and both the vapor concentrations and the isotope measurements. Lowering the
sample source relative to the outflow results in systematically heavier isotopic values in the vapor measured by
the instrument. Vapor concentrations show a similar trend, i.e. more vapor was generated for lower positions of





the sample source. These observations suggest that the hydraulic head difference directly affected the flow rate
of the liquid sample through the CWS membrane tube. Because the water is much colder than the surrounding
air as it enters the membrane chamber, it is continuously warming as it travels through the membrane tube. At
greater head gradients (and thus smaller flow rates), the sample will travel more slowly through the membrane
chamber and will warm up more. As a consequence of higher water temperatures, water can be expected to
diffuse more rapidly through the membrane and the resulting vapor can be expected to be less fractionated
relative to the liquid phase (Kendall and McDonnell, 1998), as observed in Fig. 2.

It is unknown whether the empirical linear relationships shown in Fig. 2 are generally applicable, or are specific
to each individual membrane or to the properties of the sample. Nevertheless, for this membrane and this
sample, the results indicate that changing the hydraulic head by 50 cm changes the reported isotope values by
approximately 0.12 ‰ for $\delta^{18}O$ and 0.52 ‰ for $\delta^2H$, respectively. This flow-rate artifact might become
particularly important for applications in which isotope standards and samples are drawn from sample
containers at different elevations relative to the waste outlet of the CWS (e.g. shipboard sampling). In such
cases, a vapor concentration correction relative to a reference height would have to be carried out to account for
the changes in flow rate that affects the isotopic composition in the measured water vapor. Alternatively, a
different injection system could be used to deliver a specified flow rate, independent of the position of the
source relative to the CWS. We used the Dosino for this purpose, since it functions as a high-precision syringe
pump whose delivery rate is specified by the pulse rate of the stepper motor, independent of the hydraulic head
gradient.

Because of the limited volume of each Dosino's glass cylinder (50 mL), a sample could be injected at a flow
rate of 1 mL min$^{-1}$ for a maximum of 50min. For longer injections, or to switch samples, a second Dosino had
to take over the sample delivery. The handoff between the Dosinos interrupted the sample flow to the CWS for
around 2s. This interruption was reflected in a sharp but brief increase in vapor concentrations and isotope
values, which returned back to stable values approximately 10min after the injection started (see Fig. 3 for an
example). For our application, i.e. synchronous IC measurements, we programmed a 30min injection period for
the isotope analysis. To obtain the final isotope values of a liquid sample we averaged the individual 6s
measurements reported by the WS-CRDS during the last 10min of each 30min injection period, using the first
20min to minimize any memory effects from the previous sample or from Dosino changeover. The advantage
of the Dosino-based sample injection is the very steady, pressure-independent sample injection.
**3.2 Performance of the isotope analyzer with Continuous Water Sampler (CWS)**
We quantified precision, drift coefficients and carryover effects of the isotope analyzer with CWS and Dosino-
based sample injection, using a continuous 48h laboratory experiment that alternated between three water
samples (i.e., to mimic streamwater, precipitation and a reference standard). The sample handling system was
as shown in Fig. 1, except that the precipitation collector was replaced with a 10 L bottle of nanopure water and
the streamwater sampler was replaced by a 10 L bottle of tap water. The sampling system alternated between
these two sources, and for each eighth injection it introduced an isotopically heavier secondary standard (Fiji
bottled water) (Fig. 3). The isotopic differences between Fiji bottled water and tap water were about

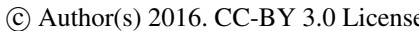



(± 1 standard error, *SE*) 4.54 ± 0.02 ‰ and 32.67 ± 0.08 ‰ for $\delta^{18}O$ and $\delta^{2}H$, respectively. The isotopic
differences between tap water and nanopure water were much smaller (0.05 ± 0.01 ‰ for $\delta^{18}O$ and
0.12 ± 0.03 ‰ for $\delta^{2}H$) because the nanopure water was generated from the same tap water by reverse osmosis.

The precisions of the isotope values, as quantified by the standard deviations of the individual 6s measurements
during the last 10min of each injection period, were better than 0.08 ‰ for $\delta^{18}O$ and 0.18 ‰ for $\delta^{2}H$. These
standard deviations imply that the standard errors of the 10min averages should be better than 0.008 ‰ and
0.018 ‰ for $\delta^{18}O$ and $\delta^{2}H$, respectively. These standard errors overestimate the repeatability of successive
measurements, however. As a measure of sample-to-sample repeatability, the standard deviations of the 10min
averages for the entire 48h experiment were 0.03 ‰ ($\delta^{18}O$) and 0.17 ‰ ($\delta^{2}H$), or better, for each of the three
water samples (excluding two outliers associated with an interruption in the sampling routine), much larger than
the calculated standard errors. Thus, the major uncertainties in the 10min averages do not arise from the
counting statistics of the instrument itself, but rather, we suspect, from sample-to-sample variability in the
performance of the vaporizer. We use these larger estimates of uncertainty (0.03 ‰ for $\delta^{18}O$ and 0.17 ‰ for
$\delta^{2}H$) in the error propagation calculations presented in Section 5.1.

Instrument drift was analyzed by linear regression of the 10min averages from the ends of each 30min injection
period. Instrument drift for $\delta^{18}O$ was statistically indistinguishable from zero for two of the three waters,
averaging (±1*SE*) -0.009±0.008, -0.009±0.006, and -0.015±0.007 ‰ day$^{-1}$ for Fiji, nanopure, and tap water,
respectively. Instrument drift for $\delta^{2}H$ was slow but statistically significant for two of the three waters, averaging
0.133±0.040, 0.084±0.016, and -0.021±0.021 ‰ day$^{-1}$ for Fiji, nanopure, and tap water, respectively. Thus the
accumulated drift over one day was typically smaller than the measurement precision for individual 10min
averages for either isotope. As explained in Section 4.2 below, substantially faster drift occurred during the
field experiment due to biofilm growth on the membrane, but could be easily measured and corrected using
regularly injected reference standards.

Between-sample memory mainly arises from small remnants of previously injected samples that remain in the
sample handling system (e.g., tubes, membrane, valves, pumps) or the analyzer itself, and are carried over to the
following analysis. We quantified the between-sample memory effect of the isotope analyzer using two
isotopically contrasting samples, Fiji water and nanopure water. The true isotopic difference was obtained from
the 7$^{th}$ (=last) injection of nanopure water, which was measured around 3h after the reference standard (Fiji),
and was thus assumed to be free of any memory effects. We calculated the memory coefficient (*X*) as a measure
of carryover effects using Gupta et al. (2009):
$$X = \frac{c_i - c_{i-1}}{c_{true} - c_{i-1}} \qquad (1)$$
where *C* denotes the isotope ratio (or the solute concentration), the indices (*i*) and (*i-1*) denote the current and
the previous injection, and (*true*) denotes the true value taken from the last value of multiple injections. The
average carryover from the Fiji bottled water to the next 30min sample was 100%·(1-X)≈0.9% for $\delta^{18}O$ and
1.2% for $\delta^{2}H$, respectively (Table 1).





**3.3 Performance of the ion chromatograph (IC)**
With the IC, a 48h laboratory experiment was carried out as well.  However, the sampling sequence differed
slightly from that of the isotope analyzer described previously: each measurement of tap water or Fiji water was
followed by two to six samples of nanopure water, which mimics precipitation samples with generally very low
solute concentrations.  Due to the low solute concentrations in the nanopure water, carryover effects can be
quantified efficiently.

Average concentrations, of the major anions and cations during the 48h experiment are reported in Table 1,
along with their absolute and relative standard deviations.  For tap water and Fiji water, relative standard
deviations were <5% for all constituents with concentrations above the limit of quantification (LOQ), indicating
that the IC measurements were stable over the 48h period.  Consequently, drift effects in the instrument were
not statistically significant ($p>0.05$) for most constituents in Fiji water and tap water.  For $Cl^-$, $NO_3^-$ and $SO_4^{2-}$ in
the Fiji water, the linear drift was statistically significant but also very slow: accumulated drift over 24h was
never much larger than the LOQ (Table 1).  Average % carryover ($100\% \cdot (1-X)$, Eq. (1)) in the nanopure water
sample, following immediately after a tap water or Fiji water sample, was $\leq 3.8$ %.
**4. Application in the field**
**4.1 Setup**
For the field experiment, the system was installed in a hut next to a small perennial stream flowing behind the
Swiss Federal Institute for Forest, Snow and Landscape Research (WSL) near Zurich, Switzerland.  The creek
drains an area mainly covered with open grassland, grain fields, and suburban residential neighbourhoods (Fig.
4).  The dominant soil type is colluvial, partly gleyic brown soil (GIS-ZH, 2016).

Stream stage, temperature and electrical conductivity were recorded in the stream every 10min using a data-
logging sonde (model DL/N 70; STS SensorTechnik Sirnach, Switzerland).  The volumetric discharge was not
gauged, but we assume that the times of the highest stream stage coincided with peak flow, and thus use both
terms synonymously. Precipitation (rainfall and snow) was measured with an unheated collector daily at
7:30am.  For a higher temporal resolution, precipitation rates at the site were estimated as the average of 10min
measurements at three nearby weather stations (Stetten, Zurich Fluntern, and Zurich Affoltern) in the
MeteoSwiss observation network.  Good agreement ($R^2 > 0.82$) was observed between measured daily
precipitation at our field site and the daily sums of the averages of the three MeteoSwiss stations, thus indicating
that the MeteoSwiss data are a reasonalble proxy for precipitation rates at the field site.  To distinguish rain and
snowfall events, air temperature was recorded near the instrument hut every 10min (Haeni, 2016; Schaub et al.,

261   2011).


A submersible pump (Eheim GmbH, Deizisau, Germany) continuously pumped streamwater at a rate of
6 L min$^{-1}$ into a through-flow bucket inside the hut.  The volume of the bucket was 10 L; thus every several
minutes the contents of the bucket were effectively exchanged.  Every 30min, water was drawn from the bucket
by the 'S Dosino' through a 1μm cellulose filter to supply the isotope analyzer, IC and autosampler (Fig. 1).





Precipitation was collected with a heated 45cm diameter funnel installed 2.5m above ground. Precipitation
flowed into a Teflon®-coated collector with a level detector that triggered at a threshold volume of 72 mL
(equaling roughly 0.5 mm of precipitation). The status of the level detector was queried before the end of each
measurement routine and a precipitation sample was drawn only if the threshold volume was exceeded. For
initial filtration of the precipitation sample, a ceramic frit filter was attached on the suction tube of the 'P
Dosino' that drew the sample from the precipitation collector. After precipitation was sampled, a peristaltic
pump emptied the precipitation collector to avoid mixing fresh and old precipitation samples. The sampling
routine was programmed to always alternate between streamwater and precipitation samples in order to obtain
enough streamwater samples during storm periods. To reduce biofilm growth on the membrane in the CWS,
copper wool was placed in the beaker from which the 'Isotope Dosinos' drew the samples. Sampling was
interrupted approximately once a week for basic maintenance (i.e., replacing the filter membranes, cleaning
Dosinos, refilling reference standards and eluent stock solutions).

Reference standards were analyzed every 3h to correct for instrument drift. Correction for drift was carried out
for the five samples between two bracketing measurements of the same reference standard:
$C_{corr} = C_{raw} + (C_{true} - \frac{C_{std,i} + C_{std,j}}{2})$                                                 (2)
with *C* denoting the solute concentration or the isotope ratio, respectively. The indices represent the corrected
value (*corr*), the current raw measurement (*raw*), the true value of the reference standard (*true*), and the
previous and successive measurements of the same reference standard (*std*) measured at time *i* and 3h later at
time *j*. For the isotope analyzer, Fiji bottled water was used as drift control, which was injected directly by one
of the 'Isotope Dosinos' (Fig. 1). The measurements of the IC were drift-corrected with another reference
standard (Evian bottled water) in the autosampler transferred directly to the IC by the 'IC Dosino'. Evian
bottled water was used as its mineral composition resembles that of streamwater more closely than Fiji bottled
water.
**4.2 Temporal high-resolution measurements of stable isotopes and major ions in precipitation and**
**streamwater**
The measurement system was deployed at the field site from 13 February 2016 to 11 March 2016 and more than
1000 streamwater and precipitation samples were analyzed for stable water isotopes and major ions. Although
the field-based measurement period covered only around 1 month, this real-time analysis system captured a
wide range of hydrological and hydrochemical conditions. Table 2 provides an overview of the eight storm
events during that period. A comparison of the aggregated precipitation data with the on-site daily
measurements from the un-heated rainfall collector indicated that precipitation during Events #1-#7 was mostly
rainfall. Snowfall occurred occasionally after 1 March, while during Event #8 most precipitation fell as snow.

We calculated the response time of streamflow as the time difference between the first detection of precipitation
and the first significant increase in streamwater level relative to the initial conditions. Typical response times
were between 0h and 2h (Table 2), suggesting an influence from the residential area in the eastern part of the
catchment. A more delayed streamflow response (4h) was observed after the snowfall event (#8), reflecting
delayed snowmelt. As illustrated by Fig. 5, a 30min sampling interval was sufficient to resolve the temporal




patterns of stable isotopes and solutes in streamflow during the rising limb of the hydrograph, even during low-
intensity precipitation periods such as Event #5.

Compared to the laboratory experiment with the isotope analyzer, during the field experiment we observed
carryover effects in the isotope measurements of up to 100%·(1-$X$)=3%, which can be explained by the copper
wool in the beaker from which the "Isotope Dosinos" drew the water samples. Despite the rinsing routine of the
beaker, the wool retained small volumes of sample from previous injections that affected the isotopic
composition in the fresh sample. Consequently, the wool was removed and the prior isotope measurements
were adjusted with $X$=97% and Eq. (1). Further, instrument drift was substantially faster during the beginning
of the field experiment due to biofilm growth in the membrane tube. For instance, during the first week,
instrument drift for raw $\delta^{18}O$ and $\delta^2H$ measurements in Fiji bottled water was statistically significant, averaging
($\pm 1SE$) -0.185±0.006 and -0.288±0.015 ‰ day$^{-1}$, respectively.

Figure 6a depicts the local meteoric water line obtained from the isotopic measurements in precipitation. The
isotopic composition of precipitation varied over a range of 14.9 ‰ in $\delta^{18}O$ and 109.4 ‰ in $\delta^2H$. A correlation
between air temperature and the isotopic composition of precipitation is evident during most storm events.
Figure 5 shows that, for instance, precipitation samples became isotopically heavier during Events #2 and #8
when air temperature increased, while the opposite behavior was observed during Events #1, #3 and #5, when
air temperature decreased. During Events #4, #6 and 7, however, the correlation with temperature was not as
distinct as during the other five events.

The isotopic composition of streamwater varied by less than half as much as that of precipitation, i.e. by 5.9 ‰
for $\delta^{18}O$ and by 43.6 ‰ for $\delta^2H$, respectively (Fig. 6b). For all eight events, the isotopic signature of pre-event
streamwater was relatively constant, averaging -10.89±0.21 ‰ for $\delta^{18}O$ and -74.88±1.40 ‰ for $\delta^2H$,
respectively (±1 standard deviation, n=8). During the events, $\delta^{18}O$ and $\delta^2H$ in streamwater changed by up to
4.54 ‰ and 34.43 ‰, respectively (Event #7).

For the IC, memory effects were negligible during the field experiment (because the sample did not make
contact with the copper wool), so the measurements were corrected only for drift effects. Solute concentrations
in precipitation and streamwater varied widely, as shown for instance in Fig. 5 for Cl$^-$ and NO$_3$$^-$. For Li$^+$, NH$_4$$^+$,
K$^+$, F$^-$ and PO$_4$$^{3-}$ in streamwater, as well as concentrations of Mg$^{2+}$ in precipitation, measured concentrations
were generally below the LOQ. NO$_3$$^-$ (as well as Ca$^{2+}$ and SO$_4$$^{2-}$, not shown) in streamwater exhibited clear
dilution patterns during all precipitation events (Fig. 5d). Concentrations of NO$_3$$^-$, Ca$^{2+}$ and SO$_4$$^{2-}$ in
precipitation during the eight events were on average (±1 standard deviation) 1.5±1.1 mg L$^{-1}$, 12.1±2.9 mg L$^{-1}$
and 0.5±0.8 mg L$^{-1}$, respectively. Solute concentrations in pre-event streamwater were on the order of (±1
standard deviation) 11.7±1.8 mg L$^{-1}$ for NO$_3$$^-$, 160.8±9.7 mg L$^{-1}$ for Ca$^{2+}$ and 21.5±3.3 mg L$^{-1}$ for SO$_4$$^{2-}$, whereas
concentrations during all events dropped to values as low as 3.73 mg L$^{-1}$ (NO$_3$$^-$), 64.6 mg L$^{-1}$ (Ca$^{2+}$) and
5.12 mg L$^{-1}$ (SO$_4$$^{2-}$). In contrast, the concentrations of Cl$^-$ (and Na$^+$, not shown) in streamwater showed dilution
patterns until Event #3, and then showed distinct enrichment patterns occurred thereafter (Fig. 5c), likely





associated with road salt wash-off. Due to possible road-salt effects on $Na^+$ and $Cl^-$, we will focus on $Ca^{2+}$, $NO_3^-$
and $SO_4^{2-}$ in the analysis below.
**5. Comparison of event-water fractions estimated from isotopes and chemical tracers**
**5.1 Hydrograph separation and uncertainty analysis**
To illustrate a potential application of high-frequency isotope and chemical measurements, here we quantify the
event-water fractions during the eight major events captured during the 1-month observation period. We used
two-component end-member mixing analysis to quantify the fractions of event water in streamflow during the
precipitation events. We applied the conventional mass balance equation (Pinder and Jones, 1969):
$F_E = \frac{Q_E}{Q_S} = \frac{C_S - C_P}{C_E - C_P}$       (3)
The fraction of event water relative to total streamflow ($F_E = Q_E/Q_S$) was calculated from the isotope values or
solute concentrations in total streamflow ($C_S$), event precipitation ($C_E$) and pre-event streamflow ($C_P$). Here, $C_P$
was obtained for each event from the average of the five streamwater samples immediately before the onset of
precipitation. The value of $C_E$ was the incremental, volume-weighted mean (McDonnell et al., 1990) of all
precipitation samples that were collected before the respective streamflow sample:
$C_{E,j} = \frac{\sum_{i=k}^{j} P_i C_i}{\sum_{i=k}^{j} P_i}$       (4)
with $P_i$ being the precipitation depth associated with the isotope value (or solute concentration) $C_i$ collected at
time $i$ since the starting time $k$ of the precipitation event.
Uncertainty in the hydrograph separation was quantified with Gaussian error propagation (Genereux, 1998),
using calculated standard errors ($SE$) arising from analytical uncertainties and the temporal variability of the
isotope values (or solute concentrations). Because $C_E$ is a volume-weighted mean, the standard error $SE_{CE}$ is
calculated with
$SE_{C_{E,j}} = \left[ \frac{\sum_{i=k}^{j} P_i (C_i - C_{E,j})^2}{(j-k) \sum_{i=k}^{j} P_i} \right]^{\frac{1}{2}}$       (5)
where $C_{E,j}$ denotes the volume-weighted mean, $C_i$ denotes the $i^{th}$ concentration that comprises that mean, and ($j$)
is the number of samples included in the volume-weighted mean. The standard error of $C_S$, $SE_{CS}$, arises from
the measurement uncertainties given in Table 1. For $SE_{CP}$, the same measurement uncertainties are applied, as
well as the temporal variability of the five measurements comprising $C_P$. The standard error of the event-water
fraction ($SE_{FE}$) can then be obtained by Gaussian error propagation:
$SE_{F_E} = \left\{ \left[ \frac{-1}{C_P - C_E} SE_{C_S} \right]^2 + \left[ \frac{C_S - C_E}{(C_P - C_E)^2} SE_{C_P} \right]^2 + \left[ \frac{C_P - C_S}{(C_P - C_E)^2} SE_{C_E} \right]^2 \right\}^{1/2}$       (6)

**5.2 Event-water fractions for eight storm events**
The varied weather conditions during the 28-day field experiment led to complex hydrologic responses,
resulting in a data set that illustrates the potential of these high-frequency measurements for hydro-chemical
analyses. Mixing analysis for two end-members, event water and pre-event water, was carried out for eight
storm events between 20 February and 8 March 2016, based on isotopic and chemical tracers. Event #8, where





precipitation fell partly as snow, was included in the analysis as river discharge and streamwater EC responded
within 4h after the onset of precipitation (Table 2). Hence, the temporal change in the snowmelt isotopic signal
due to fractionation was assumed to be negligible.

Isotope hydrograph separation (IHS) was performed using both $\delta^{18}$O and $\delta^2$H, whereas chemical hydrograph
separation (CHS) was carried out with the three constituents $Ca^{2+}$, $NO_3^-$ and $SO_4^{2-}$ ($Cl^-$ and $Na^+$, were not used
for CHS due to the influence of road salt at the site). We also performed hydrograph separation based on
streamwater EC, since several studies have used EC in lieu of chemical concentrations for hydrograph
separation, owing to the ease of obtaining continuous EC measurements (e.g., Dzikowski and Jobard, 2012;
Matsubayashi et al., 1993; Muñoz-Villers and McDonnell, 2012; Pellerin et al., 2008). As we did not measure
EC in precipitation directly, we had to estimate it empirically. For this, we used a standard conversion equation,
i.e., the pseudo-linear approach following Sposito (2008), to calculate EC in precipitation from the ionic
strength of the major cations and anions in the precipitation samples. We assume that the ion concentrations
measured by the IC account for the great majority of the ionic strength. In order to estimate the uncertainty of
this method, we also calculated the EC values in streamwater and compared them with the actual measurements
of the EC probe in the stream. The (absolute value) difference between the calculated and measured
streamwater-EC values averaged 20 µS cm$^{-1}$.

For the uncertainty analysis of the calculated event-water fractions, analytical uncertainties of the isotope
measurements were assumed to be 0.03 ‰ and 0.17 ‰ for $\delta^{18}$O and $\delta^2$H, respectively (Section 3.2, Table 1).
Relative uncertainties of the IC measurements were $0.006 \cdot C + 0.087$ mg L$^{-1}$ for $Ca^{2+}$, $0.028 \cdot C + 0.002$ mg L$^{-1}$ for
$NO_3^-$ and $0.037 \cdot C + 0.006$ mg L$^{-1}$ for $SO_4^{2-}$, respectively ((where $C$ is concentration in mg L$^{-1}$; Table 1). For the
EC values, a measurement uncertainty of 2% was assumed for the EC probe based on the specifications given
by the EC probe's manufacturer. The assumed uncertainty in the EC values in precipitation was 20 µS cm$^{-1}$, as
calculated above.

Two illustrative precipitation events, together with their hydrologic, isotopic and chemical responses in
streamwater, are shown in Figs. 7 and 8 (Events #1 and #2, respectively). During Event #1, total rainfall was
6.7 mm within 10h 40min, while 10.3 mm rain fell within 9h 40min during Event #2. Antecedent moisture
conditions, estimated by the total rainfall within 48 h and 24 h before the event, as well as initial streamwater
level, were relatively wet for Event #1 and relatively dry for Event #2 (Table 2).

For Event #1, $\delta^{18}$O and $\delta^2$H in streamwater followed the observed patterns in precipitation, i.e. streamwater
became isotopically lighter over time. Isotope hydrograph separations for this event yield maximum event-
water fractions ($F_{E,max}$) of 78±10 % and 60±14 % for $\delta^{18}$O and $\delta^2$H, respectively, similar to the results obtained
from the chemical tracers $Ca^{2+}$, $NO_3^-$ and $SO_4^{2-}$ (57±1 %, 65±2 % and 65±3 %) and EC (56±3 %, Fig. 7d and e).
The fraction of event water increased rapidly after the start of rainfall and declined continuously as stream stage
receded. A difference in response timing is evident for the chemical and isotope tracers in Fig. 7d and 7e: The
chemical tracers exhibited the strongest dilution effect during peak flow, whereas the isotope tracers showed the
largest response to the event roughly 2h later, possibly because the isotopic signature in precipitation became





lighter as the event progressed.  Consequently, if $C_S$ at the time of $Q_{max}$ were used to perform hydrograph
separation (Eq. (3)), isotope-based $F_E$-values would be substantially smaller (i.e., 43±6 % and 42±9 % for $\delta^{18}$O
and $\delta^2$H, respectively) than the $F_{E,max}$-values reported above.

During Event #2, the solutes in streamwater showed a clear dilution signal (Fig. 8c), similar to Event #1.  The
isotopic composition in streamwater, by contrast, showed only a very weak and inconsistent response to
precipitation.  For instance, $\delta^2$H in precipitation increased continuously through the event, whereas $\delta^2$H in
streamwater first decreased and then, ca. 4 h after the onset of precipitation, began to increase again.
Consequently, IHS and CHS yield substantially different interpretations for Event #2.  Maximum event-water
fractions based on CHS ranged from 68±1 % (Ca$^{2+}$) to 83±5 % (NO$^{3-}$), similar to Event #1.  In contrast, $F_{E,max}$-
values based on IHS ranged from 7±1 % to 16±3 %, indicating that pre-event water was the dominant source of
streamwater during peak flow.

How can such a large discrepancy between the event-water fractions calculated from different environmental
tracers be explained? From Fig. 5 it can be seen that precipitation was isotopically lighter than streamwater
during the six days leading up to Event #2.  Thus, the initial decrease in the $\delta^{18}$O and $\delta^2$H values in streamwater
during Event #2 suggests the release of isotopically lighter soilwater and groundwater that were recharged
during previous events.  An activation of this pre-event water storage might have been triggered by enhanced
infiltration after relatively dry antecedent moisture conditions (AMC), compared to the previous event, whereas
wet AMC would be more consistent with surface runoff generation.  This hypothesis is further supported by the
isotopic responses in streamwater during Event #5, another isotopically heavy event with dry AMC, following
earlier inputs of isotopically lighter precipitation.  In Event #5, small event-water fractions (12±1 % and 20±1 %
for $\delta^{18}$O and $\delta^2$H, respectively; Fig. S1) were again obtained, indicating that pre-event water dominated
streamflow, similarly to Event #2.  And in Event #5, just as in Event #2, the chemical tracers showed strong
dilution, leading to an overestimate of the event-water fraction (>40±2 %).  In both Event #2 and Event #5, the
chemical and isotopic data point to a large contribution from recent antecedent moisture that had not yet become
highly mineralized, rather than from either event precipitation or from older groundwater that presumably
accounted for most of the pre-event baseflow.

Figure 9 summarizes the estimated event-water fractions for all eight events, based on IHS and CHS, for two
points in time during each event: the time with the largest isotopic or chemical response (i.e., $F_{E,\ max}$) and the
time of peak flow ($Q_{max}$).  Maximum event-water fractions varied greatly across the eight events (for example,
from 16±3% to 68±14% based on $\delta^2$H, Fig. 9, Table S1 and S2).  Also, within individual events, hydrograph
separations based on different isotopic and chemical tracers differed, often by much more than their
uncertainties.  Inconsistencies between the estimated event-water fractions can be explained with the fact that
different tracers are shaped by different hydrochemical processes and flow pathways, and thus may describe
different end-members (e.g., Richey et al., 1998; Wels et al., 1991).  While stable water isotopes are considered
to be ideal conservative tracers, chemical tracers are altered by biogeochemical processes on their way through a
hydrological system.  These biogeochemical processes also vary over time, as they depend on antecedent
conditions and precipitation characteristics.  Continuous high-frequency analysis of environmental tracers can





document this temporal variability, which, in turn, helps to constrain conceptual catchment models. As
illustrated by Events #2 and #5, comparing chemical and isotopic tracers can be useful in identifying the
temporally variable contributions of different water storages in the subsurface.

Figure 9 illustrates further that for three events (#2, #5 and #8), estimated event-water fractions for the two
isotopes, $\delta^{18}O$ and $\delta^2H$, differed significantly (i.e., by more than twice their pooled uncertainties). These
differences did not follow any particular pattern, for instance, $F_E(\delta^{18}O) > F_E(\delta^2H)$ for Event #8, while $F_E(\delta^{18}O)$
$< F_E(\delta^2H)$ for Events #2 and #5. A possible explanation for such discrepancies is that the isotopic signature of
precipitation sampled at one location might not be representative of the spatially distributed precipitation that
generated the sampled streamflow (e.g., Fischer et al., 2015; Lyon et al., 2009). Alternatively, the pre-event
streamflow signature ($C_P$) may not reflect the isotopic signature of the entire pre-event water storage, but only of
the components that feed baseflow (e.g., Klaus and McDonnell, 2013). Another way of viewing this problem is
that the precipitation event may have mobilized a third pre-event water storage with unknown isotopic
composition (e.g., Tetzlaff et al., 2014). This conjecture is strongly supported by the initial shift toward
isotopically lighter streamflow early in Event #2, even though the event precipitation was isotopically heavier
than the pre-event baseflow. Event #5 also shows divergent event-water fractions between the two isotopes, and
like Event #2, it also had strongly contrasting pre-event precipitation inputs. Thus, the history of both events
suggests that pre-event storage in this catchment was isotopically heterogeneous. This observation is
unsurprising, given the pervasive heterogeneity of typical catchments, but a more detailed explanation is not
possible with our spatially limited data set. Spatially distributed measurements, such as from groundwater and
soil water storages, would help in constraining the individual end-members that contribute to streamflow (e.g.,
Hangen et al., 2001). Additional high-frequency time series of the groundwater table and soil moisture profiles
would allow for documenting the effects of antecedent wetness conditions on the response times and on the
activation of different storages at the site. Finally, a spatially distributed precipitation sampling network might
help to fully quantify the uncertainty inherent in the event-water signature.
**5.3 Variable response times of chemical and isotope tracers**
Measuring isotopes and solutes at high temporal resolution over several storm periods allows for a detailed
investigation of response times of hydrological and hydrochemical variables and their linkages to the event
characteristics. As can be seen for instance in Fig. 7, during Event #1 the timing of the largest hydrological and
hydrochemical responses did not always coincide. For only three events (i.e., #2, #4, #6) the timing of peak
flow coincided with the $F_{E, max}$ values for both chemical and isotope tracers. During Event #3, the isotope
tracers resulted in $F_{E, max}$ values 1.5h±1.0 h before peak flow. For Events #7 and #8, which were affected by
snowmelt, both tracer types showed the strongest responses up to 2.0±1.0 h earlier than the actual flow peak. In
contrast, during Event #1 the peak responses in the isotope tracers and EC came up to 2.0h±1.0 h after peak
flow.

These examples illustrate that the hydrological conditions of the stream (i.e., the stream stage or flow rate) are
not reliable proxies for the timing of the maximum event-water contribution. As a consequence, collecting
samples only during or after peak flow may result in a significant underestimation of event-water fractions. Our





498 data indicate that the time window for sample collection at our site must extend more than 3h before and after

499 peak flow in order to capture the whole range of event water dynamics.  In the case of the snowmelt Event #8,

500 the EC data suggest an even longer sampling period in order to capture unusual events such as the inflow of

501 water contaminated by road salt.

502

503 **5.4 The role of the sampling frequency for capturing hydrological and hydrochemical catchment**
504 **processes**

505 A sampling frequency can be considered optimal when the gain of information from additional measurements is

506 marginal (Kirchner et al., 2004; Neal et al., 2012).  With our high-resolution data set we can thus investigate the

507 potential of different sampling frequencies for capturing hydrological and hydrochemical catchment processes,

508 by subsampling the time series at smaller sampling frequencies, i.e. at 3-hourly, 6-hourly, 12-hourly and daily

509 intervals.  For concentrations and isotope values in streamwater, data were simply sub-sampled from the 30min

510 resolution time series to mimic grab sampling.  To mimic the effects of integrated bulk precipitation samples,

511 concentrations in precipitation were calculated from the volume-weighted averages of the 30min data over the

512 respective time interval.

513

514 Figure 10 shows that 3h sampling intervals would still be sufficient to capture the isotopic variations in

515 streamwater, including during low-intensity precipitation events.  However, the short-term variability within

516 single storm periods cannot be resolved at this lower sampling frequency.  Thus, even sampling intervals of 3h

517 can result in a significant loss of information relative to 30min sampling, and at sampling intervals of 12h or

518 longer, diurnal fluctuations and some isotopic and chemical responses to low-intensity precipitation events

519 would also be lost.  Likewise, the 6h or 12h bulk precipitation samples shown in Fig. 10 fail to reflect the large

520 isotopic variability revealed by the 30min samples.

521

522 To further illustrate the effect of lower sampling frequencies, we performed hydrograph separation with the

523 subsampled data sets, for which illustrative results are shown for the isotope tracer $\delta^2$H in Fig. 11.  With a

524 sampling interval of 3h, event-water fractions similar to those for the 30min sampling can still be obtained,

525 except for Event #3, when the 3h sampling interval captured a streamwater sample that was isotopically very

526 similar to the pre-event water.  For Events #2, #3, #5 and #7, longer sampling intervals underestimate event-

527 water fractions.  With 12h sampling intervals, IHS with $\delta^2$H yields much smaller event-water fractions for all

528 events except Event #4, and yields unrealistic results for two Events (#1, #5), as the isotopic differences

529 between the two end-members become too small.

530

531 Because the hydrologic response times in this catchment were mostly much shorter than 2h, the durations of the

532 maximum hydrochemical variations were similarly short.  Thus, sampling at longer time intervals increases the

533 risk of missing this critical peak response; if the sample is taken before or after the maximum hydrochemical

534 response, the event-water signal in streamwater ($C_S$) may be too weak, which will inevitably underestimate

535 event-water fractions, or even lead to unrealistic negative values.





**6 Concluding remarks**

This paper presents the first field hydrology application of Picarro's Continuous Water Sampler (CWS) module, which was coupled to a L2130-*i* Wavelength Scanned-Cavity ring-down Spectrometer to measure the stable water isotopes $\delta^{18}O$ and $\delta^2H$ in streamwater and precipitation at a temporal resolution of 30min. We combined this real-time isotope analysis system with a dual-channel ion chromatograph for synchronous analysis of major cations and anions. Good instrument performance and high measurement precision could be achieved during continuous 48h laboratory experiments and a 28-day deployment in the field at a small, partly urbanized catchment in central Switzerland.

Problematic issues such as sample degradation during storage and transportation, which arise in conventional sampling for catchment tracer studies, become irrelevant with the system presented here. At the same time, potential registration errors arising during the collection and handling of large numbers of water samples are avoided. Conversely, two major limitations of the coupled isotope analyser / IC system are its high cost, and the need for sufficient electrical power (around 1.7 kW), constraining its use in remote locations. However, laboratory analysis of conventionally collected grab samples is also cost-intensive, and autosamplers used in conventional sampling schemes also require a reliable energy supply (though at much lower power levels).

The results of the high-frequency analysis system were presented here to provide a proof-of-concept and an illustration of its functionality at the field, rather than to fully document the hydrological and biogeochemical processes at this field site. A more detailed interpretation would require additional measurements of soilwater and groundwater isotopes and chemistry, in order to better constrain the end-members in the mixing analysis. Nevertheless, our one-month field experiment demonstrates the marked short-term variability of several natural tracers in a small, highly dynamic watershed. The hydrograph separation exercise clearly showed that long-term, high-frequency isotopic and chemical analyses are essential for capturing the "unusual but informative" events that shed light on catchment storage and flow processes. We further showed that the right timing for capturing peak event-water contributions can easily be missed with conventional grab sampling strategies, resulting in an underestimation of the event-water fraction. In addition, the relative timing of the isotopic and chemical responses was highly variable, demonstrating the challenge of capturing the right moments with episodic snapshot campaigns or long-term monitoring with daily, weekly, or even monthly sampling intervals.

As was shown here and elsewhere (e.g., Kirchner, 2003), short-term responses of streamflow and environmental tracers may follow distinctly different patterns, which helps in constraining streamflow-generating mechanisms and quantifying short transit times. Thus, high-frequency isotopic and chemical measurements also have great potential for catchment model validation. Potential future applications of the system could include sites with rapid hydrologic responses, such as urban streams (e.g., Jarden et al., 2016; Jefferson et al., 2015; Soulsby et al., 2014), wastewater- and drinking water systems (e.g., Houhou et al., 2010; Kracht et al., 2007) or agricultural catchments with artificial drainage networks (e.g., Doppler et al., 2012; Heinz et al., 2014).





573 **Acknowledgements**

574 We thank Anton Burkhardt and the facilities staff of the Swiss Federal Institute for Forest, Snow and Landscape

575 Research (WSL) for logistical support, and Matthias Haeni from the Long-term Forest Ecosystem Research

576 Programme (LWF) at WSL for providing air temperature data. We also would like to thank Kate Dennis and

577 David Kim-Hak of Picarro Inc. (Santa Clara, CA, USA) for technical advice.





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





**Tables**

**Table 1: Average isotope values and solute concentrations, as well as standard deviations (and relative standard deviations RSD) of three water samples analyzed during two different 48h experiments with the isotope analyzer and IC, respectively. In Fiji bottled water, diluted tap water and nanopure water, concentrations of F⁻, Li⁺, K⁺, NH₄⁺ and PO₄³⁻ were mostly below the LOQ, and thus were not included in the table. The calculation of the average memory coefficient is described in the text (Eq. (1)). The uncertainties of the IC measurements were obtained by simple linear regression analysis of the average value and the standard deviation of the respective constituent.**

| | Isotope laboratory experiment | | IC laboratory experiment | | | | | |
|---|---|---|---|---|---|---|---|---|
| | $\delta^{18}O$ | $\delta^2H$ | $Na^+$ | $Mg^{2+}$ | $Ca^{2+}$ | $Cl^-$ | $NO_3^-$ | $SO_4^{2-}$ |
| Limit of quantification (LOQ) (mg L⁻¹) | - | - | 0.1 | 0.1 | 0.1 | 0.05 | 0.05 | 0.05 |
| Measurement uncertainty (‰) or (mg L⁻¹) | 0.03 | 0.17 | 0.053+ 0.005·$C$ | 0.008+ 0.006·$C$ | 0.087+ 0.009·$C$ | 0.027+ 0.003·$C$ | 0.028+ 0.002·$C$ | 0.037+ 0.006·$C$ |
| **Water sample** | **Fiji bottled water** | | **Fiji bottled water** | | | | | |
| Number of measurements | 12 | 12 | 10 | 10 | 10 | 10 | 10 | 10 |
| Average value (‰) or (mg L⁻¹) | -4.86 | -35.89 | 21.6 | 15.7 | 24.3 | 9.69 | 1.05 | 1.56 |
| Standard deviation (‰) or (mg L⁻¹) | 0.06 | 0.26 | 0.1 | 0.1 | 0.3 | 0.06 | 0.05 | 0.03 |
| RSD (%) | - | - | 0.5 | 0.4 | 1.1 | 0.60 | 4.3 | 1.80 |
| Linear drift ((‰ 24h⁻¹) or (mg L⁻¹ 24h⁻¹) | -0.009±0.008 | 0.133±0.040 | 0.129± 0.056 [a] | 0.058± 0.036 [b] | 0.093± 0.160 [c] | 0.088± 0.019 | -0.078± 0.008 | 0.045± 0.007 |
| **Water sample** | **Tap water** | | **Diluted tap water** | | | | | |
| Number of measurements | 34 | 34 | 17 | 18 | 18 | 18 | 18 | 18 |
| Average value (‰) or (mg L⁻¹) | -9.40 | -68.55 | 10.9 | 34.4 | 133.2 | 12.41 | 4.96 | 17.29 |
| Standard deviation (‰) or (mg L⁻¹) | 0.03 | 0.12 | 0.1 | 0.2 | 1.3 | 0.057 | 0.03 | 0.14 |
| RSD (%) | - | - | 0.7 | 0.6 | 1.0 | 0.5 | 0.7 | 0.8 |
| **Water sample** | **Nanopure water** | | **Nanopure water (last sample)** | | | | | |
| Number of measurements | 43 | 43 | 27 | 27 | 27 | 27 | 27 | 27 |
| Average value (‰) or (mg L⁻¹) | -9.44 | -68.67 | <LOQ | 0.1 | 0.6 | <LOQ | <LOQ | 0.09 |
| Standard deviation (‰) or (mg L⁻¹) | 0.02 | 0.18 | 0.02 | 0.003 | 0.1 | 0.03 | 0.02 | 0.05 |
| Carryover (%) | 0.9 | 1.2 | 2.8 | 3.3 | 3.8 | 2.1 | 1.9 | 2.3 |

[a] $p > 0.05$
[b] $p > 0.15$
[c] $p > 0.50$





**Table 2:** Characteristics of precipitation events and antecedent moisture conditions during the field experiment. Initial stream stage is used here as a proxy for initial discharge.

| Event | Start of event | Total precipitation (mm) | Total precipitation until peak flow (mm) | Response time (h) | 48h antecedent precipitation (mm) | 24h antecedent precipitation (mm) | Initial stream stage (cm) |
|---|---|---|---|---|---|---|---|
| #1 | 14 February 2016 10:30 | 6.7 | 5.1 | 01:40 | 8.5 | 2.9 | 0.44 |
| #2 | 20 February 2016 12:30 | 10.3 | 9.2 | 00:00 | 1.3 | 0.0 | 0.36 |
| #3 | 23 February 2016 07:00 | 5.0 | 4.8 | 00:00 | 0.2 | 0.2 | 0.37 |
| #4 | 24 February 2016 15:30 | 15.3 | 11.1 | 01:00 | 5.2 | 3.3 | 0.41 |
| #5 | 28 February 2016 05:50 | 10.6 | 2.9 | 01:10 | 0.0 | 0.0 | 0.38 |
| #6 | 02 March 2016 12:30 | 6.0 | 6.0 | 01:50 | 11.9 | 2.0 | 0.46 |
| #7 | 05 March 2016 05:20 | 9.4 | 8.6 | 01:30 | 4.3 | 0.9 | 0.45 |
| #8 | 07 March 2016 21:00 | 6.4 | 6.4 | 04:00 | 1.9 | 0.0 | 0.45 |





**Figures**

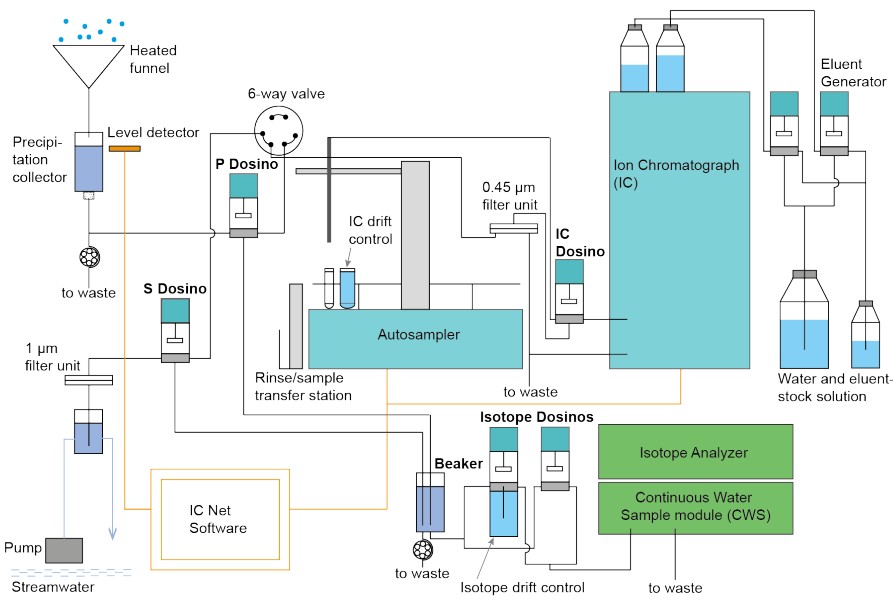

**Figure 1: Schematic overview of the coupled isotope analyzer / IC- system for the collection and measurement of streamwater and precipitation samples. Components of the sample distribution and the IC are shown in blue color, while the isotope analyzer with**
5  **CWS is shown in green color.**

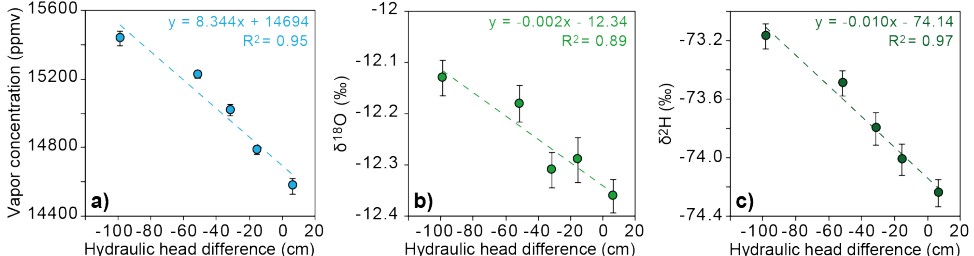

**Figure 2: Measured vapor concentrations (panel a)), and isotope ratios (panels b) and c)) of a single water sample (nanopure**
10  **water) as a function of the hydraulic head difference between the water level in the sample bottle and the waste outlet. Negative values of the hydraulic head difference indicate that the sample source was located below the waste outlet of the CWS.**



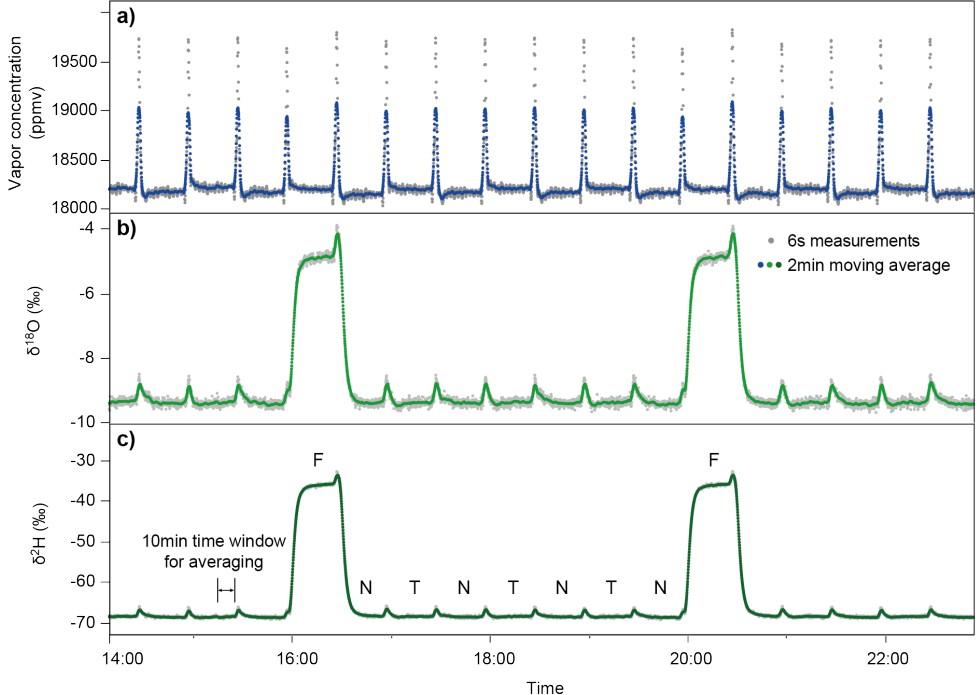

**Figure 3: Nine hour excerpt showing vapor concentrations (panel a)) and isotope measurements (panels b) and c)) in tap water (T), nanopure water (N) and Fiji bottled water (F) during the 48h laboratory experiment. Samples were injected alternately with two Dosinos for 30min each at a flow rate of 1 mL min⁻¹.**





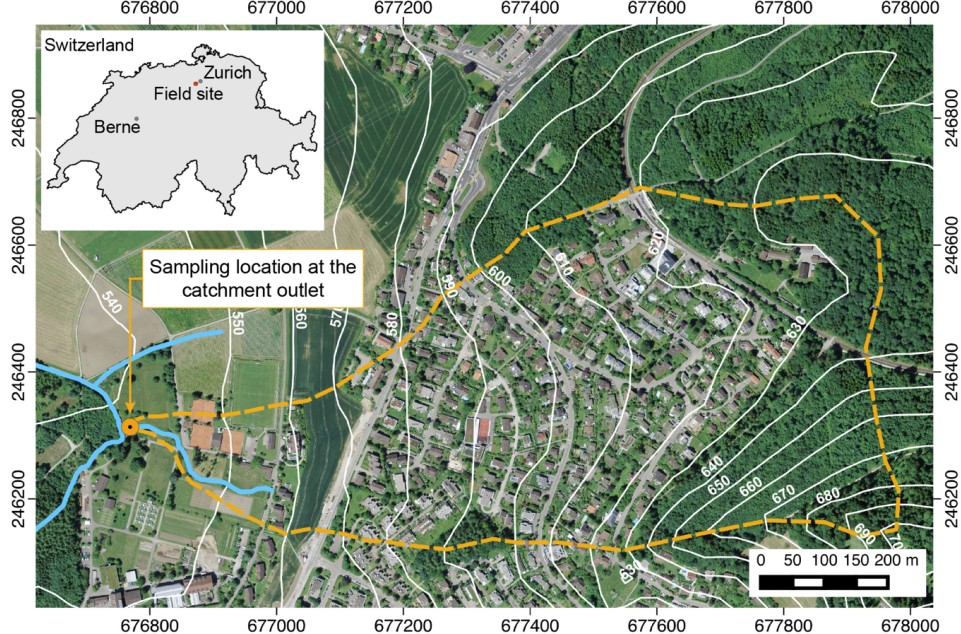

**Figure 4: Location of the field site at a small creek on the property of the Swiss Federal Institute for Forest, Snow and Landscape Research (WSL) near Zurich, Switzerland. Catchment boundaries are approximate.**




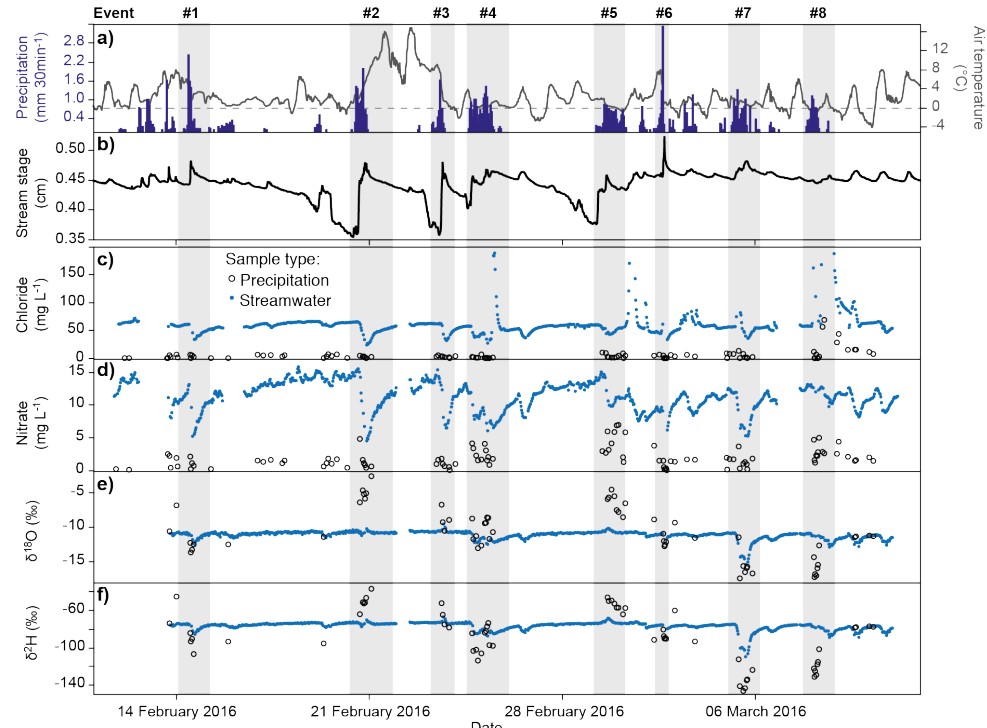

**Figure 5:** Time series of precipitation, air temperature (a) and stream stage (b) at the field site during the four-week study period. Panels c) and d) show the chloride and nitrate concentrations, whereas panels e) and f) show the isotopic compositions. Streamwater samples are shown by blue dots and precipitation samples are shown by open circles. Vertical grey bars indicate the periods of the eight precipitation events used for hydrograph separation.





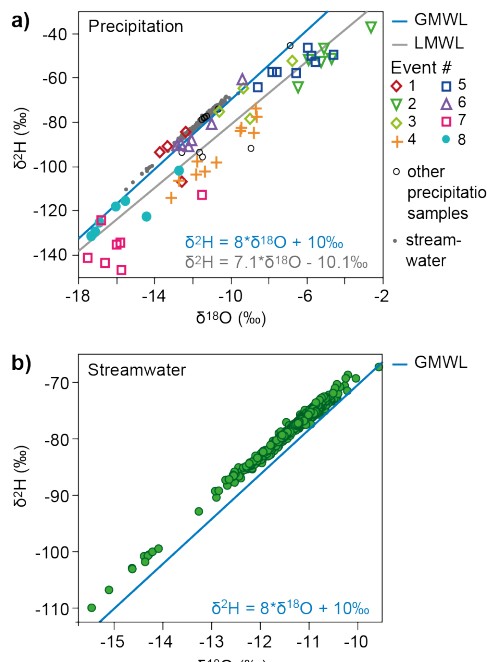

**Figure 6: Dual-isotope plot of all δ¹⁸O and δ²H values measured in precipitation (a) and streamwater (b) during the field experiment. Streamwater samples are also plotted in grey in the upper panel for comparison (note the difference in scales). The global meteoric water line (GMWL) and the linear fit to the precipitation data (local meteoric water line, LMWL) are shown in blue and in grey, respectively.**





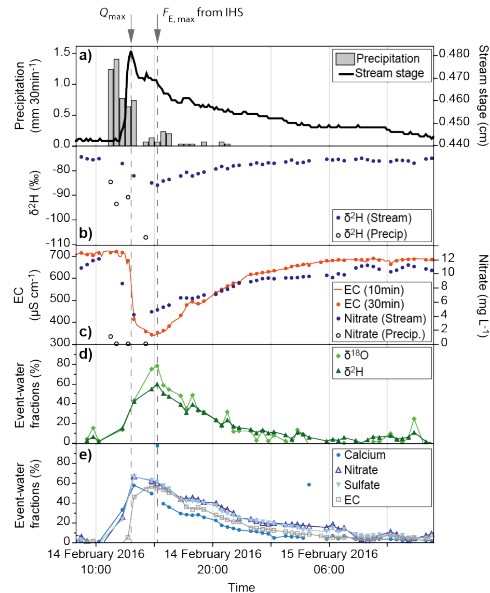

**Figure 7: Precipitation Event #1 together with the hydrologic (a), isotopic (b) and chemical (c) responses in streamwater. Panels d) and e) show the fractions of event-water based on isotopic and chemical hydrograph separation, respectively, which are similar for both types of tracers. However, the timing of the maximum event-water fraction ($F_{E,max}$) differs, i.e. the isotopes indicate the**
5   **largest contribution of event water around 2h after the flood peak ($Q_{max}$) was reached. In panel e), gaps in the $F_E$ time series based on calcium concentrations are due to measurement outliers.**





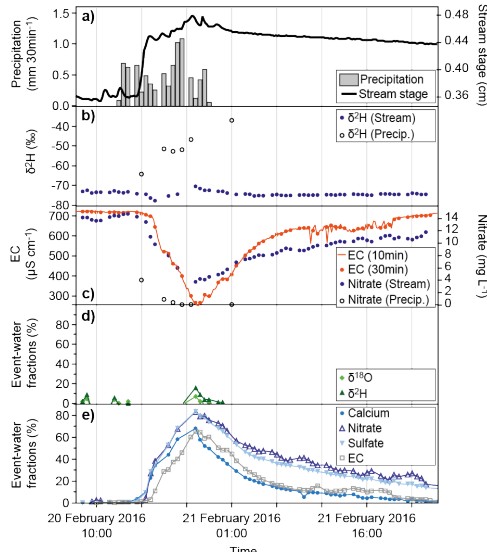

**Figure 8: Precipitation Event #2 and the hydrologic, isotopic and chemical responses in streamwater. Panels d) and e) show the fractions of event water ($F_E$) based on isotopic and chemical hydrograph separation. Chemical tracers greatly exaggerate the event-water fraction.**





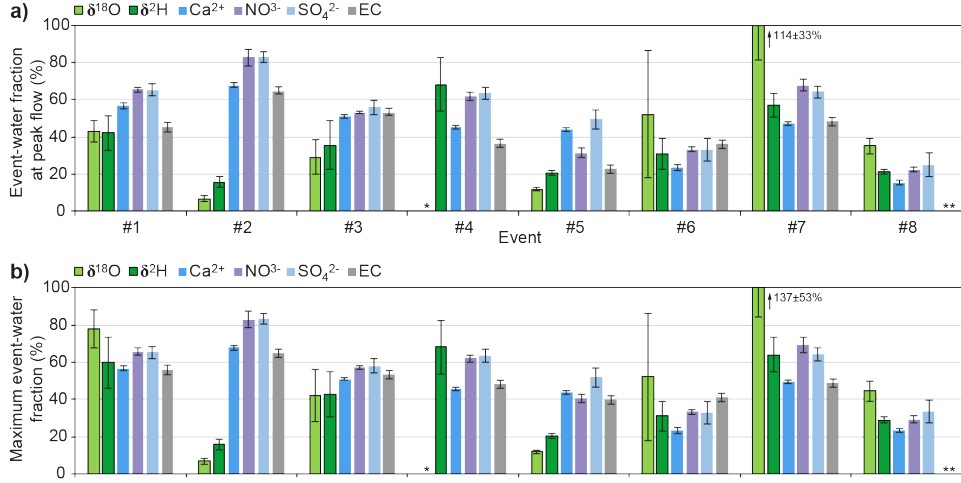

Figure 9: Event-water fractions ($F_E$) based on isotopic and chemical hydrograph separation for eight storm events. Panel a) shows $F_E$ during peak flow, and panel b) shows the maximum event-water fractions ($F_{E,max}$) of each event. Unrealistic $F_E$ and $F_{E,max}$ values were obtained for Event #4 based on $\delta^{18}O$ because the isotopic signatures in precipitation and pre-event streamwater were too similar (*). For Event #8, wash-off of road salt resulted in unrealistic $F_E$ and $F_{E,max}$ values based on EC, i.e. -95±7% (**).





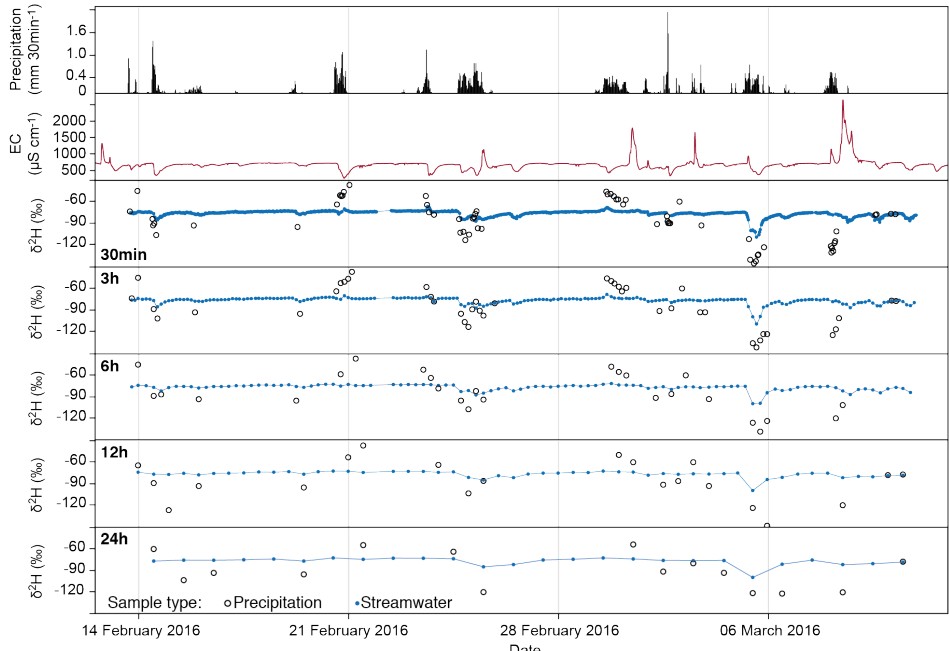

**Figure 10: Time series of precipitation and streamwater EC (at 10min temporal resolution), as well as δ²H values in streamwater and precipitation at sampling intervals of 30min, 3h, 6h, 12h and 24h.**





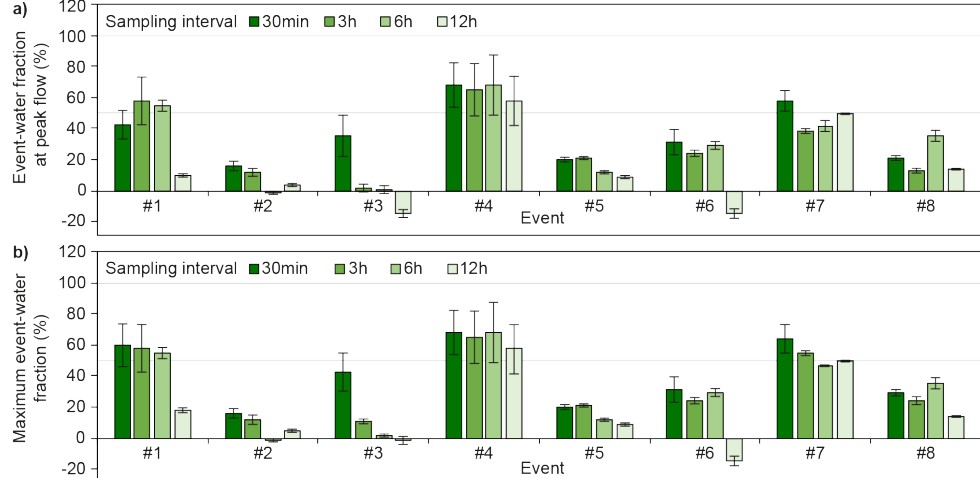

**Figure 11: Event-water fractions at peak flow (a) and maximum event-water fractions (b) based on δ²H measurements at sampling intervals of 30min, 3h, 6h and 12h. With lower sampling frequencies, the event-water fractions are often underestimated or become even unrealistic, as the likelihood increases that the point of largest δ²H variations in streamflow will be missed.**