# Peer review of "A lab in the field: High-frequency analysis of water quality"

_Hydrology and Earth System Sciences, 2016_

## Referee Comment (RC1) · N. Munksgaard (Referee) · 23 Nov 2016

General comment: This paper describes the development and performance of an advanced high-frequency analyser of both water isotopes and major ions in stream flow and precipitation. The authors provide a thorough account of the instrument design and operation and an assessment of analytical performance. The integration of the many components represents a significant engineering effort. The instrumentations analytical data quality is impressive, in particular the precision of isotope analysis. The instrumentation is described as a 'lab in the field' and high quality data can certainly be produced in real time during extended deployments. However, I question whether it can be described as a true field instrument considering its limited portability, multi

component complexity (Fig. 1) and relatively high requirements for power and shelter. The field deployment described was in an outbuilding of the research institute, presumably with mains power but it is unclear what environmental conditions the instrument was exposed to (e.g. range of temperature fluctuations). A photograph of the actual setup used would be a useful addition. The last 1/3 of the paper (Section 5 - comparison of event water fractions) is concerned with the interpretation of the month long field trial. This section seems somewhat too long given that the main aim of the paper (as per the introduction) is the description of the development and field trial of the instrument (these are adequately described through sections 1-4). Moreover, the interpretation provided in section 5 is somewhat hampered by lack of data on important potential contributions to stream flow (soil and groundwater) as only two endmembers (base flow and precipitation) are considered. This limitation is clearly acknowledged by the authors. A substantial shortening of section 5 should be considered, although a shortened comparison of event-water fractions based on isotopes and ions should be retained as it provides a very good demonstration of the value of high frequency isotope measurement compared to the more traditional use of solute tracers in discrete samples taken at longer intervals. The manuscript is clearly written and the conclusions are sound and well supported by the data presented.

Specific comments: 2. Methodology: For the laboratory based tests the analysis system was not calibrated as only relative isotope values were required - however, it is not clear if full calibration to the VSMOW scale or only drift correction was performed in the field tests – please expand on this (P 8 L280). Figures 5 and 6 display actual field data, e.g. in Fig. 5 data is shown relative to GMWL and LMWL so this comparison would require that full calibration was performed. One of the limitations / uncertainties in the calculation of event-water fractions is (as stated by the authors) the precise definition of end member compositions. As the isotope composition (O and H) is often regarded as the most reliable tracer of event water it could be argued that the highest possibly frequency of isotope measurement of both stream and precipitation water should be prioritised. In this regard it seems illogical that the measurement of

isotope compositions was limited to 30 min intervals in order to synchronise data with the IC measurements which required this amount of time. This is particularly the case when a continuous water isotope instrument was used, wouldn't it be more beneficial to fully utilise its capability to perform truly continuous analysis, especially for precipitation which can vary significantly over much shorter time intervals than 30 min? At 30 min intervals, a 'conventional' CRDS instrument with a sequential injection/evaporation cycle could equally well have been used (apart from possible maintenance requirements). A similar auto sampling system was used for the IC and presumably this required regular maintenance.

3.1 Optimisation of sample injection. . . The explanation provided for the delta dependence on hydraulic head difference (P 5 L 152 and Fig. 2) may not be the full explanation. The Picarro CWS uses a high air flow rate (possibly ≈500 mL/min as I recall) and this has the effect that the vapour generated is not in isotopic equilibrium with the liquid water flowing inside the ePTFE tube. The vapour is significantly depleted in d18O and d2H compared to equilibrium values due to the faster diffusion through the membrane of the light isotopologues compared to the heavy. This effect does not matter much as long as air and water flows and temperatures are kept constant for both sampling and calibration standards. However, the large fractionation effect probably tends to make the system prone to artefacts such as an altered pump rate. The use of a relatively cheap peristaltic water pump as opposed to the CWS supplied diaphragm pump would provide improved flow and lift characteristics (as does the expensive Dosimo pumps used in this study). P 6 L 213. Drift is attributed to biofilm growth, was this growth assumed or actually observed? Possibly temperature drift (inlet air and water) was also a factor in the field deployment? Was the instrumentation exposed to outdoor temperature fluctuations or was temperature regulated indoors? P 6 L 217. How constant was the memory effect? If relatively constant, a data correction could be applied. Presumably it would be a function of analysis time (washout effect)

4. Application in the field: P 9 L 323. It is unclear what 'opposite behaviour' means, a

number of interpretations are possible. . .please clarify

5. Comparison of even-water. . . P 11 L 407: consider using 'precipitation' instead of 'moisture' P12 L 435, 442: I agree this is likely - as has been shown by Tweed et al. 2016 (Hydrol. Process. 30, 648–660 2016). It would also be relevant to cite this publication in the Introduction as it appears to be the first study using continuous real-time isotope monitoring to trace contributions to storm flow. P 13 L 463 and Figure 9: Event #7 results (»100% event water) needs commenting on. . . P 13 L 466 onwards: Seems unlikely there was spatial variation in rainfall in such a small catchment ($\approx$ 0.5 km2). Possibly O and H isotope compositions of other contributing (but not measured) water sources (groundwater, soil water, over land flow) had variable d18O / d2H relations (d-excess values). Since these were not accounted for the simple two-component fraction calculations based on O and H could differ. P 14 L 523: Somewhat ironically this section concludes that 3-hourly sampling would have been sufficient to capture the stream water events and would result in similar calculated event-water fractions. Consequently the stream data could have been monitored using a conventional auto sampler at relatively low cost and with much simpler operation. While this may not be the case in all storm water events it may be the case that it is precipitation monitoring that will benefits the most by continuous isotope instrumentation due to the very rapid (minutes) changes that can occur in precipitation isotope values.

Concluding remarks: The limitations regarding field deployability of the system (my general comments) and possible options for improvements in this regard could be expanded upon.

Table 2: stream stage unit must be m (not cm) Figure 3: Note that the recorded water vapour concentrations ($\approx$ 18,200 ppm) corresponds to a T of $\approx$16.2 oC supporting the explanation given by the authors that water warmed up beyond the 15oC setting of the diffusion cell of the CWS Figure 6: GMWL and LMWL require references (especially the latter) Figure 9: Error bars and their large variation between events need mention in legend and main text Figure 10, 11: Legends should clarify that the 3, 6, 12, 24 hour

'sampling intervals' were derived by re-sampling of the 30 min data

---

## Referee Comment (RC2) · Anonymous Referee #2 · 19 Dec 2016

General comment: In their work 'A lab in the field: High frequency analysis of water quality and stable isotopes in streamwater and precipitation' von Freyberg et al. present the set up and proof-of-concept of a new automatic sampling system for high-resolution measurements of stable water isotopes and stream solutes. The analytical components mainly consist of a laser spectroscopy system for stable water isotope analysis and an ion chromatograph for the measurement of anions and cations. Both instruments are joined by a sophisticated peripheral set up. Particular emphasis has been put into the reduction of carry over effects during operation when switching between the two water sources sampled.

The paper is clearly structured and easy to read. The system's set up is impressive

and the authors made great effort with a proof-of-concept. The precision achieved for both systems is excellent. I recommend to accept the paper for publication. However, I see some aspects of their set up that should be acknowledged in a revised version:

(1) The idea of a lab in the field is nice, but in fact the system requires a proper housing and full power connection. The system is far from being mobile and field deployable. To my understanding it is still a lab in a house (which is located in vicinity to a catchment).

(2) Automatic wet chemistry instrument set ups are routinely operated by water provisioning services (e.g. in sewage treatment works) and by larger state environmental agencies or research facilities for surface water monitoring. Such systems are expensive (and therefore hardly used in basic research projects), but in place. See reviews from Gray et al. 2006, Environmental Chemistry, http://dx.doi.org/10.1071/EN05059 or Bende-Michl and Hairsine 2010, Journal of Environmental Monitoring, DOI: 10.1039/B910156J.

(3) Even though the authors achieved a reasonable high sampling frequency of 30 min this set up does not allow to capture the potentially very high temporal change of stable water isotopes in precipitation. This short term variation (that is missed by aggregating to 30 min composite samples or by sampling only the last precipitation sequence of a 30 min period) might be responsible for some of the differences found for the chemical- and isotope-based hydrograph separations. For high temporal resolution application see results in Pangle et al. 2013, already cited in the paper, or Moerman et al. 2013, Earth and Planetary Science Letters, http://dx.doi.org/10.1016/j.epsl.2013.03.014. I do not fully understand why the two sampling systems for isotope and chemical compositions are not operated independently, but parallel. In a synchronized, parallel set up, the systems could sample stable water isotopes independently in the order of minutes utilizing the full power of the continuous water sampler while (composite or grab) samples for the ion chromatograph are sampled in 30 min resolution.

(4) The current set up is limited to two water sources. Given the set up and limitation of 30 min sampling frequency the system is only partly extendable with regard to sampling additional sources such as groundwater, tributaries or nearby standing water bodies. The sampling of additional sources, however, is needed to partially resolve the differences in the various hydrograph separations outlined in Chapter 5.

Specific comments.

L16: Missing space between value and unit, check throughout the manuscript

L47 'much longer much smaller' – I do not get what you intend to say.

L48-52 There are two new papers out by Aubert and Breuer 2016, PLoS ONE, doi:10.1371/journal.pone.0153138 and Aubert et al. 2016, Scientific Reports, DOI: 10.1038/srep31536, that explicitly show the value of high resolution measurements of nitrate, might be worth considering here (or at least to have a look at).

L101 You sampled an internal standard every 3 h (see Line 280). So why did you not use these standards to correct absolute isotope values?

L246 An installation in a hut with power supply is not an installation in the field as the title of the paper indicates. At least not to my understanding of a field deployable system. I suggest to revise the title.

L257 I do not agree that the correlation of daily precipitation is good and therefore allows to derive subdaily precipitation patterns. We know how variable precipitation can be in space. As almost 20% of the variance of daily sums cannot be explained by the correlation, I can only speculate on the potential differences in hourly or even higher frequency precipitation rates. For future application of the system, I strongly suggest to add an independent met station to the field lab, which is negligible in costs compared to all the other instruments and peripherals used to set up this lab.

L264 Is there any reason to use a large bucket of 10 L rather than a substantially smaller container to sample from? Or utilize an even larger container to produce composite rather than grab samples. Chapter 5 As the paper is mainly a description of the

technical set up, I wonder a bit on the extensive evaluation of the event/pre-event water contribution in this chapter. While I see this a very interesting aspect of the utilization of the system in a fully operational application, I think this section can be reduced for this more technical paper. In fact, the discussed potentially missing end member to better understand the observations (groundwater, soil water, spatially variable precipitation) reflects the limitation of the current set up where only two water sources can be sampled in high temporal resolution (see also general comment, bullet point (4)).

L423-483 The difference of the isotope and chemical tracer derived hydrograph separation are enormous. The dissimilarity of results is so large, that the overall applicability of the approach seems to be questionable. As the authors stress, it goes beyond the scope of this paper to resolve the issue and further end members need to be identified to explain the observations. Thus, the discussion remains at least partly speculative. The current set up of the system does not allow to sample more sources in high resolution. To this end, the system described in the paper is limited to investigate more simple mixing processes of two end members, rather than complex mixing processes typical for catchments. Part of the observation might be due to the non-conservativeness of the chemical tracers. At least for NO3- I doubt its feasibility to be used as a real tracer, particularly in highly biological reactive top soils. Another potential explanation is variable source areas and their connectivity to the stream, with source areas being switched on/off during events.

L444 revise expression: moisture cannot be mineralized

Chapter 5.4 This section misses a real discussion. There are at least a few papers that look into the effect of sampling frequency on hydrograph separation or mean transit time estimation that could be discussed here. Stockinger e tal. 2014, http://dx.doi.org/10.1016/j.jhydrol.2016.08.007; Timbe et al. 2015 doi:10.5194/hess-19-1153-2015; Birkel et al. 2012 DOI: 10.1002/hyp.8210; Inamdar et al. 2013 DOI: 10.1002/wrcr.20158.

L514 I do not agree that you can conclude from Fig 3 that a 3 h sampling frequency would be sufficient. May be you can conclude this from Fig 11. However, in L516 you note yourself that even intervals of 3 h can results in a significant loss of information.

L566 Despite the technical achievement and the effect of the high frequency sampling, the authors could stress even more the highly uncertain results of their hydrograph separation (even though this should not be the major focus of the paper). Combining the results of Fig 9 and Fig 11 I conclude that whatever model you have, it could easily be verified by an 'appropriate' selection of solutes and their sampling frequency, as the uncertainty to derive a 'true' hydrograph separation is very large.

Chapter 6 (or elsewhere in the discussion). After I read the conclusions, I wonder why the authors have decided to include an ion chromatography system that limits really high resolution measurements and therefore also limits sampling further water sources. At least for some of the solutes measured, ion selective probes or UV hyperspectral photometry sensors (reviewed by Rode et al. and already included in the paper) could be used as an alternative analytical system with substantially lower costs as well.

---

## Author Comment (AC1) · 10 Jan 2017

**Response to the interactive comment of Reviewer #1 on**

 **"A lab in the field: high-frequency analysis of water quality and stable isotopes in streamwater and precipitation"** *by* **Jana von Freyberg, Bjørn Studer and J.W. Kirchner**

*Comments of the reviewer are shown in italics.*

Responses from the authors are presented in regular font below each comment.  Citations from the manuscript are in Times New Roman, changes of the text in the underlined.

1.  *General comment: This paper describes the development and performance of an advanced high-frequency analyser of both water isotopes and major ions in stream flow and precipitation. The authors provide a thorough account of the instrument design and operation and an assessment of analytical performance. The integration of the many components represents a significant engineering effort. The instrumentations analytical data quality is impressive, in particular the precision of isotope analysis. The instrumentation is described as a 'lab in the field' and high quality data can certainly be produced in real time during extended deployments. However, I question whether it can be described as a true field instrument considering its limited portability, multicomponent complexity (Fig. 1) and relatively high requirements for power and shelter. The field deployment described was in an outbuilding of the research institute, presumably with mains power but it is unclear what environmental conditions the instrument was exposed to (e.g. range of temperature fluctuations).*

    General response: we thank Dr. Niels Munksgaard for his thoughtful comments on our paper, and for developing his diffusion-based water isotope sampling system, which was the inspiration for the continuous water sampler that was used in our work.

    The terminology "lab in the field" indicates the use of analytical instruments that are usually employed in a laboratory environment (i.e, isotope analyzer Picarro L2130-i, ion chromatograph).  Thus, the title makes clear that a small-scale laboratory was set-up at a field site to allow for real-time water sample analysis at high-precision.  Nowhere did we claim that this was a "lab in a box" or a "field instrument".  We point out in the manuscript that substantial infrastructure (accessibility, power access) is required to run the lab continuously.  A completely remote system powered by solar panels would be impractical because of the high power requirements.

    All instruments were located in a small, wooden hut without additional insulation.  The outside air temperatures, which are also shown in Figure 5a, were on average 2°C and ranged between -4 and 17°C.  In the hut, temperatures were generally around 14°C and ranged between 7 and 23°C, because of the heat produced by the instruments.  Humidity in the hut was around 40%.

2.  *A photograph of the actual setup used would be a useful addition.*

    A photo of the setup will be added to Fig.1.

3.  *The last 1/3 of the paper (Section 5 - comparison of event water fractions) is concerned with the interpretation of the month-long field trial. This section seems somewhat too long given that the main aim of the paper (as per the introduction) is the description of the development and field*

trial of the instrument (these are adequately described through sections 1-4). Moreover, the interpretation provided in section 5 is somewhat hampered by lack of data on important potential contributions to stream flow (soil and groundwater) as only two endmembers (base flow and precipitation) are considered. This limitation is clearly acknowledged by the authors. A substantial shortening of section 5 should be considered, although a shortened comparison of event-water fractions based on isotopes and ions should be retained as it provides a very good demonstration of the value of high frequency isotope measurement compared to the more traditional use of solute tracers in discrete samples taken at longer intervals. The manuscript is clearly written and the conclusions are sound and well supported by the data presented._

Sections 5.1 and 5.2 comprise only around ¼ of the whole manuscript (1905 of 7724 words), which includes the theory of HS, a detailed description of two contrasting (and thus interesting) events, and the conceptual description of runoff generation mechanisms at the site.  We believe that these topics help the reader to understand how high-frequency time series of water isotopes and major ions can be used to study catchment hydrological processes.  We will look for ways to streamline this section, while retaining its value for the reader (which is also recognized by the reviewer).

**Specific comments:**

4. *2. Methodology: For the laboratory based tests the analysis system was not calibrated as only relative isotope values were required - however, it is not clear if full calibration to the VSMOW scale or only drift correction was performed in the field tests – please expand on this (P 8 L280).*

   The results reported in the manuscript were based only on drift correction using secondary isotope standards (Fiji and Evian bottled water) rather than calibration to the VSMOW scale. However, we have now calibrated these secondary standards to primary IAEA standards (SLAP, VSMOW, GISP), and field results in the final version of the paper will be both drift-corrected and calibrated to the VSMOW scale.

5. *Figures 5 and 6 display actual field data, e.g. in Fig. 5 data is shown relative to GMWL and LMWL so this comparison would require that full calibration was performed.*

   As explained in our response to Comment 4., the final version of the paper will report values that are both drift-corrected and calibrated to the VSMOW scale, and thus will be comparable to the GMWL.  Figures 5 to 8 will be updated accordingly.

6. *One of the limitations / uncertainties in the calculation of event-water fractions is (as stated by the authors) the precise definition of end member compositions. As the isotope composition (O and H) is often regarded as the most reliable tracer of event water it could be argued that the highest possibly frequency of isotope measurement of both stream and precipitation water should be prioritised. In this regard, it seems illogical that the measurement of isotope compositions was limited to 30 min intervals in order to synchronise data with the IC measurements which required this amount of time. This is particularly the case when a continuous water isotope instrument was used, wouldn't it be more beneficial to fully utilise its capability to perform truly continuous analysis, especially for precipitation which can vary significantly over much shorter time intervals than 30 min? At 30 min intervals, a 'conventional' CRDS instrument with a sequential injection/evaporation cycle could equally well have been used (apart from possible maintenance requirements). A similar auto sampling system was used for the IC and presumably this required regular maintenance.*

A "truly continuous" analysis is not possible with this instrument (or any other isotope analyzer that we know of) due to the memory effects within the instrument itself. As Fig. 3 clearly shows, significant memory effects persist within the sampler and analyzer for at least 10 minutes after the previous sample injection. Thus, sampling at 10-minute intervals (for example) would produce measurements that are strongly affected by the previous samples, as the isotopic signals overprint each other.

These memory effects might not be so problematic if we were only analyzing streamflow which changes relatively gradually. But instead, the sampling system must switch between rainwater and stream water during precipitation events, and thus sample carryover effects could lead to substantial distortions in subsequent calculations (such as hydrograph separations). We have chosen the 30-minute sampling interval in the interests of minimizing carryover effects, even if (say) 20-minute sampling would be potentially achievable.

7. *3.1 Optimisation of sample injection. . . The explanation provided for the delta dependence on hydraulic head difference (P 5 L 152 and Fig. 2) may not be the full explanation. The Picarro CWS uses a high air flow rate (possibly ≈500 mL/min as I recall) and this has the effect that the vapour generated is not in isotopic equilibrium with the liquid water flowing inside the ePTFE tube. The vapour is significantly depleted in d18O and d2H compared to equilibrium values due to the faster diffusion through the membrane of the light isotopologues compared to the heavy. This effect does not matter much as long as air and water flows and temperatures are kept constant for both sampling and calibration standards. However, the large fractionation effect probably tends to make the system prone to artefacts such as an altered pump rate. The use of a relatively cheap peristaltic water pump as opposed to the CWS supplied diaphragm pump would provide improved flow and lift characteristics (as does the expensive Dosimo pumps used in this study).*

Dr. Munksgaard's comment provides additional background to the explanation that we already give in the manuscript, starting on line 153: "Because the water is much colder than the surrounding air as it enters the membrane chamber, it is continuously warming as it travels through the membrane tube. At greater head gradients (and thus smaller flow rates), the sample will travel more slowly through the membrane chamber and will warm up more. As a consequence of higher water temperatures, water can be expected to diffuse more rapidly through the membrane and the resulting vapor can be expected to be less fractionated relative to the liquid phase (Kendall and McDonnell, 1998), as observed in Fig. 2." We can of course modify this explanation to include the additional point that evaporation through the membrane is highly fractionating (which we thought was sufficiently obvious that it did not need to be said).

A cheap peristaltic water pump might also be an improvement over the diaphragm pump that is supplied with the CWS, but would present its own maintenance issues (aging and wear of pump tubing, for example). We used the Dosino pumps, despite their higher cost, because they provide direct control over fluid flow rates.

8. *P 6 L 213. Drift is attributed to biofilm growth, was this growth assumed or actually observed?*

During prior tests at this field site the membrane was removed and a biofilm could indeed be observed. We can modify the text to make this clear.

9. *Possibly temperature drift (inlet air and water) was also a factor in the field deployment? Was the instrumentation exposed to outdoor temperature fluctuations or was temperature regulated indoors?*

The hut was not temperature controlled and thus the isotope analyzer was exposed to some temperature variations. The instruments inside the hut produced heat and thus, average temperature in the hut was about 12°C higher than outside. The variations of outside air temperature were clearly reflected inside the hut and in the membrane temperature of the CWS (streamwater temperature mirrors air temperature as well). Nevertheless, there was no long-term temperature trend that correlates with the drift observed in vapor concentrations (towards lower values). Except for diurnal patterns, air temperature and water temperature were rather stable - except for a warm period between 20 and 24 February 2016 when outside air temperature reached 17°C (Fig.5a). Inside the hut up to 23°C were reached during that period. This warm period did not manifest itself either in the vapor concentrations or in the isotope data of the routinely measured drift control, because the inlet air and water temperatures are both regulated using Peltier thermoelectric controllers. Once the air and water enter the membrane chamber, however, the flow rate of the water determines how much it is heated by the (much warmer) air. Thus, the temperature of the water at the membrane itself depends on the water flow rate, which is why we regulate the flow rate using the Dosino dosing pumps.

10. *P 6 L 217. How constant was the memory effect? If relatively constant, a data correction could be applied. Presumably it would be a function of analysis time (washout effect).*

To assess the stability of the memory effect during the 48-hour experiment, we calculated the percent carry-over for each Nanopure water sample injected immediately after Fiji water (when the isotopic difference between the two samples is the largest). We obtained rather stable values of percent carry-over that were (average±standard deviation) 1.25±0.35% and 0.89±0.44% for $\delta^2H$ and $\delta^{18}O$, respectively. No statistically significant trend could be observed in the percent carry-over during the 48-hour experiment.

As mentioned already in P9 L323 in the first version of the manuscript, we applied a correction for memory effects by using

$$C_{true,i} = \frac{C_i - (1-X) \cdot C_{i-1}}{X}$$

where $C_{true,i}$ is the true value, $C_i$ is the measured value, $C_{i-1}$ is the value of the immediately previous injection and X is the memory coefficient. We will keep this correction procedure in the revised version of the manuscript.

*4. Application in the field: P 9 L 323. It is unclear what 'opposite behaviour' means, a number of interpretations are possible. . .please clarify*

We have clarified this statement: "Figure 5 shows that, for instance, precipitation samples became isotopically heavier during Events #2 and #8 when air temperature increased, while the precipitation samples became isotopically lighter during Events #1, #3 and #5, when air temperature decreased."

11. *5. Comparison of event-water. . . P 11 L 407: consider using 'precipitation' instead of 'moisture'*

As we also consider initial discharge as a proxy for antecedent moisture conditions rather than antecedent precipitation per se, we would prefer not to use the term "antecedent precipitation" conditions here.

12. *P12 L 435, 442: I agree this is likely - as has been shown by Tweed et al. 2016 (Hydrol. Process. 30, 648–660 2016). It would also be relevant to cite this publication in the Introduction as it*

*appears to be the first study using continuous real- time isotope monitoring to trace contributions to storm flow.*

We will include this reference in the revised manuscript.

13. *P 13 L 463 and Figure 9: Event #7 results (»100% event water) needs commenting on. . .*

For Event #7, the large calculated event-water fractions (and standard errors) can be explained with the first $\delta^{18}$O measurement of this event, which was isotopically very similar to the pre-event water signature ($C_E$=-11.5‰, $C_P$=-10.9‰). The incremental, volume-weighted mean of the event-water end member was thus isotopically heavier than the streamwater end member, resulting in a smaller difference to the pre-event water end member signature (Eq. 3). Precipitation samples after this first, less-$\delta^{18}$O-depleted sample had an average $\delta^{18}$O value of - 16.4±0.69‰ (±1 standard deviation, n=6). For $\delta^2$H, such a strong effect did not occur and we could obtain reasonable isotope-based hydrograph separation results similar to the chemical hydrograph separation.

14. *P 13 L 466 onwards: Seems unlikely there was spatial variation in rainfall in such a small catchment (≈ 0.5 km2). Possibly O and H isotope compositions of other contributing (but not measured) water sources (groundwater, soil water, over land flow) had variable d18O / d2H relations (d-excess values). Since these were not accounted for, the simple two-component fraction calculations based on O and H could differ.*

We have moved this possible explanation to the end of this section and included the comment of the reviewer accordingly:

"Another way of viewing this problem is that the precipitation event may have mobilized a third pre-event water storage with unknown isotopic composition (e.g., Tetzlaff et al., 2014). It is further possible that the $\delta^{18}$O- $\delta^2$H relations (d-excess) of contributing water sources (groundwater, soil water, overland flow) were variable over time, resulting in different event-water fractions based on both isotopes."

15. *P 14 L 523: Somewhat ironically this section concludes that 3-hourly sampling would have been sufficient to capture the stream water events and would result in similar calculated event-water fractions. Consequently, the stream data could have been monitored using a conventional auto sampler at relatively low cost and with much simpler operation. While this may not be the case in all storm water events it may be the case that it is precipitation monitoring that will benefits the most by continuous isotope instrumentation due to the very rapid (minutes) changes that can occur in precipitation isotope values.*

We thank the reviewer for this comment, which we will implement into the revised manuscript: "Additionally, the rapid changes observed in precipitation isotopic composition (Fig. 6 and 5) suggests that high-frequency measurements are crucial for adequately represent the signature of the event-water end member."

As for the question of whether 3-hourly sampling could be done by conventional autosamplers: of course it is possible to use autosamplers at any sampling frequency, but higher sampling frequencies will necessarily entail more frequent field visits and greater numbers of bottles to be handled in the lab (with the associated quality control issues).

16. *Concluding remarks: The limitations regarding field deployability of the system (my general comments) and possible options for improvements in this regard could be expanded upon.*

The manuscript is quite explicit about the limitations of the analysis system because of the complexity of the instrumentation and its space requirements. The power requirement for the whole analysis system can only be estimated from the specifications given by the manufacturers. Based on these information, the number presented in the manuscript (around 1.7kW) considers the maximum power requirement of all instruments, for instance during warm-up. During steady-state operation we expect this number to be much smaller, however, we did not measure the power consumption directly, and thus do not know the exact number. Instead of presenting the maximum power requirement of all instruments, in the future version of the manuscript we will emphasize that line power would be optimal to allow for continuous instrument operation. Other alternatives, such a free-standing solar power system or a generator, would be possible, but these would be expensive and have their own reliability and maintenance issues.

Regarding possible options for improvements of the system regarding its field deployment we want to point out that the isotope analyzer with CWS was already optimized for "field applications" by the manufacturer. In contrast, the IC is a typical laboratory experiment that was not used in such an environment before. We are not instrument design engineers and thus will refrain from making specific recommendations for improvements beyond those we have tested ourselves.

17. *Table 2: stream stage unit must be m (not cm)*

    We will change that.

18. *Figure 3: Note that the recorded water vapour concentrations (≈ 18,200 ppm) corresponds to a T of ≈16.2 °C supporting the explanation given by the authors that water warmed up beyond the 15°C setting of the diffusion cell of the CWS.*

    We thank the reviewer for pointing this out.

19. Figure 6: GMWL and LMWL require references (especially the latter)

    We included following reference for GMWL: Gat J, Mook WG, H.A.J. M. Environmental Isotopes in the Hydrological Cycle: Principles and Applications: International Atomic Energy Agency; 2001. In Fig. 6, we have changed the term "LMWL" to "Linear fit".

20. *Figure 9: Error bars and their large variation between events need mention in legend and main text*

    The error bars of the IHS are larger than those of the CHS because of the larger temporal variability of the isotope values in precipitation; that is, isotope values in precipitation vary by much more that the analytical uncertainty of the instrument.

    We will add this information into the main text and into the caption of Fig. 9: "The larger uncertainties of the IHS results compared to CHS can be explained with the large temporal variability of the isotope values in precipitation, which substantially exceeds analytical uncertainty during most events."

21. *Figure 10, 11: Legends should clarify that the 3, 6, 12, 24 hour 'sampling intervals' were derived by re-sampling of the 30min data.*

We will include this information in the figure captions.

---

## Author Comment (AC2) · 10 Jan 2017

**Response to the interactive comment of Reviewer #2 on**

 **"A lab in the field: high-frequency analysis of water quality and stable isotopes in streamwater and precipitation"** *by* **Jana von Freyberg, Bjørn Studer and J.W. Kirchner**

*Comments of the reviewer are shown in italics.*

Responses from the authors are presented in regular font below each comment.  Citations from the manuscript are in Times New Roman, changes of the text in the underlined.

General comment: In their work 'A lab in the field: High frequency analysis of water quality and stable isotopes in streamwater and precipitation' von Freyberg et al. present the set up and proof-of-concept of a new automatic sampling system for high- resolution measurements of stable water isotopes and stream solutes. The analytical components mainly consist of a laser spectroscopy system for stable water isotope analysis and an ion chromatograph for the measurement of anions and cations. Both instruments are joined by a sophisticated peripheral set up. Particular emphasis has been put into the reduction of carry over effects during operation when switching between the two water sources sampled.

The paper is clearly structured and easy to read. The system's set up is impressive and the authors made great effort with a proof-of-concept. The precision achieved for both systems is excellent. I recommend to accept the paper for publication. However, I see some aspects of their set up that should be acknowledged in a revised version:

1.  *The idea of a lab in the field is nice, but in fact the system requires a proper housing and full power connection. The system is far from being mobile and field deployable. To my understanding it is still a lab in a house (which is located in vicinity to a catchment).*

    With the title of the manuscript we emphasize that a small-scale laboratory was set up at a field site next to a stream to allow for real-time, high-precision analysis of water samples.  We do not claim in the manuscript that the analysis system is mobile or field deployable.  All instruments were housed in a wooden hut, which was not temperature regulated, and required an area of only around $3m^2$.  Thus, the analysis system could potentially be set up at any other location with road access and sufficient power supply.  In the revised manuscript, we will include a photograph of the analysis system in the field to better illustrate the compact setup in the hut.

2.  *Automatic wet chemistry instrument set ups are routinely operated by water provisioning services (e.g. in sewage treatment works) and by larger state environmental agencies or research facilities for surface water monitoring. Such systems are expensive (and therefore hardly used in basic research projects), but in place. See reviews from Gray et al. 2006, Environmental Chemistry, http://dx.doi.org/10.1071/EN05059 or Bende-Michl and Hairsine 2010, Journal of Environmental Monitoring, DOI: 10.1039/B910156J.*

    We thank the reviewer for these additional references, which we can implement into the revised manuscript.

3.  *Even though the authors achieved a reasonable high sampling frequency of 30 min this set up does not allow to capture the potentially very high temporal change of stable water isotopes in*

*precipitation. This short term variation (that is missed by aggregating to 30 min composite samples or by sampling only the last precipitation sequence of a 30 min period) might be responsible for some of the differences found for the chemical- and isotope-based hydrograph separations. For high temporal resolution application see results in Pangle et al. 2013, already cited in the paper, or Moerman et al. 2013, Earth and Planetary Science Letters, http://dx.doi.org/10.1016/j.epsl.2013.03.014. I do not fully understand why the two sampling systems for isotope and chemical compositions are not operated independently, but parallel. In a synchronized, parallel set up, the systems could sample stable water isotopes independently in the order of minutes utilizing the full power of the continuous water sampler while (composite or grab) samples for the ion chromatograph are sampled in 30 min resolution.*

We agree that a higher sampling frequency would be desirable to resolve the isotopic variability in precipitation, however, our system was developed to measure many precipitation events in order to identify changes short-term catchment hydrological processes over time periods of weeks and months.  In the study mentioned by the reviewer, Pangle et al. (2013) only sampled one event at 34min intervals.  Thus, our study provides a more detailed insight into the variability of precipitation isotopes simply because more events were captured.  Moerman et al. (2013) also sampled only one event at 1-4min intervals.  This study also acknowledges that more events would have to be sampled at high frequencies to confirm the representativeness of the monitored event.

4. *The current set up is limited to two water sources. Given the set up and limitation of 30 min sampling frequency the system is only partly extendable with regard to sampling additional sources such as groundwater, tributaries or nearby standing water bodies. The sampling of additional sources, however, is needed to partially resolve the differences in the various hydrograph separations outlined in Chapter 5.*

We understand this point, and of course, given the sufficient resources, one could always install another isotope-IC system to measure other water sources.  We clearly acknowledge this situation at the end of section 5.2, by pointing out that the age distribution of the pre-event water cannot be fully quantified with our data set, and thus, additional end-members would have to be determined: "Thus, the history of both events suggests that pre-event storage in this catchment was isotopically heterogeneous.  This observation is unsurprising, given the pervasive heterogeneity of typical catchments, but a more detailed explanation is not possible with our spatially limited data set.  Spatially distributed measurements, such as from groundwater and soil water storages, would help in constraining the individual end-members that contribute to streamflow (e.g., Hangen et al., 2001).  Additional high-frequency time series of the groundwater table and soil moisture profiles would allow for documenting the effects of antecedent wetness conditions on the response times and on the activation of different storages at the site."

**Specific comments**

5. *L16: Missing space between value and unit, check throughout the manuscript*

We will change that.

6. *L47 'much longer much smaller' – I do not get what you intend to say.*

This was a typesetting mistake. We will correct that. "As pointed out by Kirchner et al. (2004), sampling at intervals much longer than the hydrological response times of a catchment may result in significant losses of information. "

7. *L48-52 There are two new papers out by Aubert and Breuer 2016, PLoS ONE, doi:10.1371/journal.pone.0153138 and Aubert et al. 2016, Scientific Reports, DOI: 10.1038/srep31536, that explicitly show the value of high resolution measurements of nitrate, might be worth considering here (or at least to have a look at).*

We thank the reviewer for this suggestion and will consider implementation of the references into the revised manuscript.

8. *L101 You sampled an internal standard every 3 h (see Line 280). So why did you not use these standards to correct absolute isotope values?*

The results presented in the original manuscript used only the factory calibration of the isotope analyzer. We have now used the internal standard to calibrate to reference standards, and will present the calibrated values in the revised manuscript, along with a more detailed description of our calibration and drift correction procedures.

9. *L246 An installation in a hut with power supply is not an installation in the field as the title of the paper indicates. At least not to my understanding of a field deployable system. I suggest to revise the title.*

We believe that the title concisely and accurately describes the contents of the manuscript. Our paper describes what is literally a laboratory in the field (with, as in most laboratories, power and communications and protection from weather). The point of the paper is to examine the relative merits of putting the laboratory out in the field, rather than transferring samples from the field back to a central laboratory far removed from the field site. Nowhere do we claim that the instrumentation package that we describe is a "lab in a box" or a "field deployable system" in the sense suggested by the reviewer. Instead, the manuscript quite explicitly describes the level of infrastructure that is required, and the tradeoffs involved vis-a-vis more conventional field sampling approaches.

10. *L257 I do not agree that the correlation of daily precipitation is good and therefore allows to derive subdaily precipitation patterns. We know how variable precipitation can be in space. As almost 20% of the variance of daily sums cannot be explained by the correlation, I can only speculate on the potential differences in hourly or even higher frequency precipitation rates. For future application of the system, I strongly suggest to add an independent met station to the field lab, which is negligible in costs compared to all the other instruments and peripherals used to set up this lab.*

In contrast to weather conditions in summer, when local convective storms cause highly variable, intermittent precipitation rates over time and space, the weather in northern Switzerland during winter is mainly characterized by large-scale frontal events. This results in longer, less variable and strongly autocorrelated precipitation events (Molnar and Burlando, 2008), which are likely to be captured by several meteorological stations in close vicinity of our field site. Unfortunately, the only rain gauge at our field site was not heated, and thus did not capture precipitation rates during snowfall periods. For this reason, we rely on data from the MeteoSwiss network, which used heated rain gauges. In order to account for the uncertainty pointed out by the reviewer, in the future version of the manuscript we will utilize hourly precipitation rates from the MeteoSwiss CombiPrecip dataset. This dataset is based on a geostatistical combination of radar estimates ($1km^2$-resolution) and rain-gauge measurements, to reduce uncertainties due to spatial variability of precipitation patterns.

Reference: Molnar P, Burlando P. Variability in the scale properties of high-resolution precipitation data in the Alpine climate of Switzerland. Water Resour Res. 2008; 44(10).

**11.** *L264 Is there any reason to use a large bucket of 10 L rather than a substantially smaller container to sample from? Or utilize an even larger container to produce composite rather than grab samples.*

We used a long and narrow bucket with a siphon at the outlet. Fresh streamwater continuously flowed in from the top of the bucket and left the bucket through the siphon outlet at the bottom. Thus, sediment and organic material settled down and a less turbid streamwater sample could be retrieved from the upper part of the bucket. This configuration helped to slow down the clogging of the filters substantially.

**12.** *Chapter 5 As the paper is mainly a description of the technical set up, I wonder a bit on the extensive evaluation of the event/pre-event water contribution in this chapter. While I see this a very interesting aspect of the utilization of the system in a fully operational application, I think this section can be reduced for this more technical paper. In fact, the discussed potentially missing end member to better understand the observations (groundwater, soil water, spatially variable precipitation) reflects the limitation of the current set up where only two water sources can be sampled in high temporal resolution (see also general comment, bullet point (4)).*

We agree with the reviewer and try to will shorten Chapter 5 in the revised manuscript.

**13.** *L423-483 The difference of the isotope and chemical tracer derived hydrograph separation are enormous. The dissimilarity of results is so large, that the overall applicability of the approach seems to be questionable. As the authors stress, it goes beyond the scope of this paper to resolve the issue and further end members need to be identified to explain the observations. Thus, the discussion remains at least partly speculative. The current set up of the system does not allow to sample more sources in high resolution. To this end, the system described in the paper is limited to investigate more simple mixing processes of two end members, rather than complex mixing processes typical for catchments. Part of the observation might be due to the non-conservativeness of the chemical tracers. At least for NO3- I doubt its feasibility to be used as a real tracer, particularly in highly biological reactive top soils. Another potential explanation is variable source areas and their connectivity to the stream, with source areas being switched on/off during events.*

The point of this section of the paper is precisely to demonstrate that reactive chemical tracers (including electrical conductivity) are unreliable for separating event and pre-event water in the hydrograph. It is widely understood that this can be problematic, but our data clearly show how problematic this indeed can be. We make this point because we are frequently asked to review manuscripts that use electrical conductivity for hydrograph separation, apparently unaware of the problems that this approach poses. Our point is explicitly not that we have identified all the end-members that one would need to sample to more fully characterize the different sources of "old water" within the catchment.

**14.** *L444 revise expression: moisture cannot be mineralized*

We will change that.

**15.** *Chapter 5.4 This section misses a real discussion. There are at least a few papers that look into the effect of sampling frequency on hydrograph separation or mean transit time estimation that*

*could be discussed here. Stockinger e tal. 2014, http://dx.doi.org/10.1016/j.jhydrol.2016.08.007; Timbe et al. 2015 doi:10.5194/hess- 19-1153-2015; Birkel et al. 2012 DOI: 10.1002/hyp.8210; Inamdar et al. 2013 DOI: 10.1002/wrcr.20158.*

We thank the reviewer for the suggested references, which we will include into the discussion section of chapter 5.4:

"Thus, sampling at longer time intervals increases the risk of missing this critical peak response; if the sample is taken before or after the maximum hydrochemical response, the event-water signal in streamwater ($C_S$) may be too weak, which will inevitably underestimate event-water fractions, or even lead to unrealistic negative values. Capturing the short-term responses of environmental tracers also helps in better quantifying transit time distributions (e.g., Birkel et al., 2012; Stockinger et al., 2016; Timbe et al., 2015) and in constraining concentration-discharge models (e.g., Stelzer and Likens, 2006; Jones et al., 2012)."

16. *L514 I do not agree that you can conclude from Fig 3 that a 3 h sampling frequency would be sufficient. Maybe you can conclude this from Fig 11. However, in L516 you note yourself that even intervals of 3 h can results in a significant loss of information.*

   We will clarify the statement: "Figure 10 shows that 3h sampling intervals would still be sufficient to capture the major isotopic responses in streamwater, including during low-intensity precipitation events. However, there are also several storm periods (e.g., Events #7 and #8) during which the short-term variability cannot be resolved at this lower sampling frequency."

17. *L566 Despite the technical achievement and the effect of the high frequency sampling, the authors could stress even more the highly uncertain results of their hydrograph separation (even though this should not be the major focus of the paper). Combining the results of Fig 9 and Fig 11 I conclude that whatever model you have, it could easily be verified by an 'appropriate' selection of solutes and their sampling frequency, as the uncertainty to derive a 'true' hydrograph separation is very large.*

   A 'true' separation of the hydrograph into event and pre-event water was carried out by using high-frequency measurements of stable water isotopes, which are considered to be ideal conservative tracers. As pointed out in the manuscript, large uncertainties can be explained by the variability of the isotopic signal in precipitation (i.e., isotope values in precipitation vary by much more that the analytical uncertainty of the instrument).

   Hydrograph separation based on solute data might have smaller uncertainties (due to smaller variability of solute concentrations in precipitation), however, event-water fractions can be largely over- or underestimated. As we show for Events #1 and #2, for instance, the results are not consistent, and thus the selection of 'appropriate solutes and their sampling frequencies' is not at all straight forward. The key point is that one should not be free to separate the hydrograph onto event and pre-event water based on reactive tracers, since they give results that are inconsistent with the isotopes, which are nearly ideal tracers.

18. *Chapter 6 (or elsewhere in the discussion). After I read the conclusions, I wonder why the authors have decided to include an ion chromatography system that limits really high resolution measurements and therefore also limits sampling further water sources. At least for some of the solutes measured, ion selective probes or UV hyperspectral photometry sensors (reviewed by Rode et al. and already included in the paper) could be used as an alternative analytical system with substantially lower costs as well.*

We agree that ion selective probes or UV hyperspectral photometry sensors are cheaper but measurement stability and accuracy are also substantial issues with such instruments. Of course, there are many alternatives of how an on-line system such as ours could be set up, however, each one has advantages and disadvantages.

---

## Author Response (AR1)

Zurich, 8 February 2017

Dear Prof. Markus Weiler,

Please find attached the revised version of our manuscript entitled "A lab in the field: High-frequency analysis of water quality and stable isotopes in streamwater and precipitation" (hess-2016-585).

We addressed all issues raised by the reviewers (our detailed responses to the reviewers are already posted). In particular, following the reviewer's suggestion, we have compressed section 5, for instance by merging sections 5.3 and 5.4 into one section. We further added sub-headings to section 5.2 to better structure this part of the revised manuscript and to make it easier for the reader to follow the train of thought.

Further, we included more references to previous studies to link our research to other potential approaches. Among others, we reference a review paper by Rode et al. (2016) that presents a comprehensive overview of the most recent applications of high-frequency measurements of isotopes and solutes in hydrology. Unfortunately, we did not find any oceanography studies where high-frequency isotope measurements are used (or could be used), however, we hope that our paper may arouse interest within the ocean research community.

We highly appreciated the thoughtful comments of you and the two reviewers, which helped to improve our manuscript. We believe that the revised version of our manuscript illustrates more clearly the novelty of our analysis system and describes in more detail the application of high-frequency measurements of stable water isotopes and major ions in catchment studies.

Thank you for considering our revised manuscript.

Best regards,

Jana von Freyberg, Bjørn Studer and James W. Kirchner

[revised manuscript text omitted]

**Figure 7: Precipitation Event #1 together with the a) hydrologic , b) isotopic  and c) chemical  responses in streamwater. Panels d) and e) show the fractions of event-water based on isotopic and chemical hydrograph separation, respectively, which are similar for both types of tracers.  However, the timing of the maximum event-water fraction ($F_{E,max}$) differs, with the isotopes indicate the largest contribution of event water around 32h after the  peak flow($Q_{max}$) was reached.  In panel e), gaps in the $F_E$ time series based on calcium concentrations are due to measurement outliers.**

[Figure]

**Figure 8: Precipitation Event #2 and the a) hydrologic, b) isotopic and c) chemical responses in streamwater.  Panels d) and e) show the fractions of event water ($F_E$) based on isotopic and chemical hydrograph separation.  Chemical tracers greatly exaggerate the event-water fraction.**

[Figure]

[Figure]

**Figure 9: Event-water fractions ($F_E$) based on isotopic and chemical hydrograph separation for eight storm events. Panel a) shows $F_E$ during peak flow, and panel b) shows the maximum event-water fractions ($F_{E,max}$) of each event. Unrealistic $F_E$ and $F_{E,max}$ values based on $\delta^{18}O$ were obtained for Event #4  because the isotopic signatures in precipitation and pre-event streamwater were too similar (\*). For Event #8, wash-off of road salt resulted in unrealistic $F_E$ and $F_{E,max}$ values based on EC, i.e. -96±6% and -95±6% (\*\*), respectively. The larger uncertainties of the IHS results compared to CHS can be explained with the large temporal variability of the isotope values in precipitation, which substantially exceeds analytical uncertainty during most events.**

[Figure]

Figure 10: Time series of precipitation, stream stage and streamwater EC, (at 10min temporal resolution), as well as δ²H values in streamwater and precipitation at sampling intervals of 30 min, 3 h, 6 h, 12 h and 24 h. Streamwater isotope values at 3 h – 24 h temporal resolution were obtained by sub-sampling from the 30 min time series. To mimic the effects of integrated bulk precipitation samples, isotope values in precipitation were calculated from volume-weighted averaging the 30 min data over the corresponding time intervals. Vertical grey bars indicate the periods of the eight precipitation events used for hydrograph separation.

[Figure]

**Figure 11:** aximum event-water fractions  at sampling intervals of 30 min, 3 h, 6 h and 12 h based on a) δ²H and b) EC.  With lower sampling frequencies, the event-water fractions are often underestimated or become even unrealistic, as the likelihood increases that the point of largest δ²H or EC variations in streamflow will be missed (Streamwater δ²H and EC time series were subsampled at 3-hourly, 6-hourly, 12-hourly and daily intervals; concentrations of integrated bulk precipitation samples were calculated from the volume-weighted averages over the respective time interval. For Event #8, wash-off of road salt resulted in unrealistic $F_{E,max}$ values based on EC (*).).

**A lab in the field: High-frequency analysis of water quality and stable isotopes in streamwater and precipitation**

Jana von Freyberg[1, 2], Bjørn Studer[1], James W. Kirchner[1, 2]

[1] Department of Environmental Systems Science, ETH Zurich, Zurich, Switzerland

[2] Swiss Federal Research Institute WSL, Birmensdorf, Switzerland

*Correspondence to*: Jana von Freyberg (jana.vonfreyberg@usys.ethz.ch)

[Figure]

**Figure S1: Precipitation Event #5 together with the hydrologic (a), isotopic (b) and chemical (c) responses in streamwater. Panels d) and e) show the fractions of event-water based on isotopic hydrograph separation  and chemical hydrograph separation , respectively, which are different for both types of tracers: While the  isotope tracers yields event-water fractions smaller than 2%,  chemical tracers estimated much larger event-water fractions of more than 40%.**

**Table S 1: End-members and event-water fractions during peak flow.**

| - | $\delta^{18}$O (‰) | $\delta^2$H (‰) | $Ca^{2+}$ (mg L$^{-1}$) | $NO_3^-$ (mg L$^{-1}$) | $SO_4^{2-}$ (mg L$^{-1}$) | EC (µS cm$^{-1}$) |
|---|---|---|---|---|---|---|
| **Event** | Pre-event-water end member ($C_P$) ±$SE_{CP}$ | | | | | |
| **#1** | -11.16±0.08 | -77.22±0.29 | 162.68±1.60 | 11.04±0.24 | 20.06±1.25 | 710.20±14.57 |
| **#2** | -10.81±0.07 | -75.50±0.29 | 166.58±1.59 | 13.62±0.15 | 23.98±1.48 | 718.00±14.38 |
| **#3** | -10.85±0.03 | -75.55±0.18 | 166.28±1.62 | 12.84±0.10 | 23.77±1.47 | 712.20±14.27 |
| **#4** | -10.92±0.06 | -75.78±0.34 | 158.00±1.52 | 11.53±0.21 | 21.95±1.36 | 669.80±13.92 |
| **#5** | -10.93±0.03 | -76.26±0.20 | 163.97±1.66 | 12.94±0.18 | 24.46±1.51 | 700.60±14.02 |
| **#6** | -11.36±0.05 | -79.13±0.26 | 142.07±1.56 | 8.50±0.15 | 15.20±0.99 | 586.00±12.15 |
| **#7** | -11.09±0.03 | -77.61±0.18 | 163.61±1.57 | 11.52±0.12 | 21.05±1.30 | 695.80±13.99 |
| **#8** | -11.16±0.03 | -77.96±0.19 | 167.56±1.59 | 12.14±0.16 | 22.34±1.38 | 708.60±14.23 |
| **Event** | Event-water end member ($C_E$) ±$SE_{CE}$ at peak flow | | | | | |
| **#1** | -13.21±0.36 | -91.34±2.34 | 14.78±4.32 | 0.55±0.28 | 0.16±0.08 | 64.28±26.86 |
| **#2** | -5.25±0.41 | -55.01±3.16 | 14.55±2.74 | 1.41±0.64 | 1.07±0.60 | 14.05±20.65 |
| **#3** | -8.37±0.74 | -62.05±4.08 | 17.35±1.26 | 0.48±0.13 | 0.07±0.04 | 26.41±20.29 |
| **#4** | -10.99±0.55 | -98.84±4.37 | 9.26±1.87 | 1.94±0.30 | 0.05±0.04 | 4.94±20.04 |
| **#5** | -5.78±0.10 | -47.64±1.58 | 14.36±1.05 | 3.95±0.63 | 2.60±0.66 | 12.04±20.08 |
| **#6** | -12.04±0.23 | -88.96±1.54 | 6.03±2.04 | 0.27±0.09 | 0.05±0.04 | 8.90±20.29 |
| **#7** | -14.36±1.19 | -133.59±7.28 | 11.75±2.16 | 1.52±0.50 | 0.09±0.05 | 21.99±21.07 |
| **#8** | -15.87±0.53 | -125.34±3.00 | 10.48±2.39 | 2.46±0.39 | 0.18±0.14 | 17.35±20.79 |
| **Event** | Streamwater end member ($C_S$) ±$SE_{CS}$ at peak flow | | | | | |
| **#1** | -11.43±0.03 | -79.39±0.17 | 114.90±1.11 | 8.60±0.04 | 15.72±0.98 | 414.00±8.28 |
| **#2** | -10.39±0.03 | -72.41±0.17 | 64.60±0.66 | 3.73±0.03 | 5.12±0.34 | 304.00±6.08 |
| **#3** | -9.65±0.03 | -68.88±0.17 | 96.38±0.94 | 7.36±0.04 | 13.63±0.85 | 363.00±7.26 |
| **#4** | -12.65±0.03 | -89.72±0.17 | 101.97±0.99 | 6.56±0.04 | 11.10±0.70 | 428.00±8.56 |
| **#5** | -10.32±0.03 | -70.36±0.17 | 98.23±0.96 | 9.94±0.04 | 13.56±0.85 | 442.00±8.84 |
| **#6** | -11.77±0.03 | -82.40±0.17 | 107.28±1.04 | 5.48±0.04 | 9.57±0.61 | 431.00±8.62 |
| **#7** | -15.89±0.03 | -113.99±0.17 | 88.22±0.87 | 4.62±0.04 | 7.98±0.52 | 380.00±7.60 |
| **#8** | -13.20±0.03 | -92.12±0.17 | 139.13±1.33 | 9.68±0.04 | 16.80±1.05 | 1369.00±27.38 |
| | Event-water fraction $F_E$ ±$SE$ (%) at peak flow | | | | | |
| **Event** | $\delta^{18}$O | $\delta^2$H | $Ca^{2+}$ | $NO_3^-$ | $SO_4^{2-}$ | EC |
| **#1** | 13.36±4.28 | 15.40±3.33 | 32.31±1.41 | 23.30±1.89 | 21.84±6.95 | 45.86±2.60 |
| **#2** | 7.64±1.33 | 15.09±2.75 | 67.08±1.33 | 80.98±4.28 | 82.33±2.87 | 58.81±2.10 |
| **#3** | 48.66±14.64 | 49.36±14.99 | 46.94±0.95 | 44.30±0.72 | 42.77±5.05 | 50.92±2.11 |
| **#4** | -[a] | 60.45±11.50 | 37.67±1.04 | 51.86±1.99 | 49.53±4.48 | 36.37±2.15 |
| **#5** | 11.77±0.82 | 20.59±1.40 | 43.94±0.95 | 33.39±2.72 | 49.87±5.42 | 37.56±2.11 |
| **#6** | 60.14±21.00 | 33.30±5.78 | 25.58±1.21 | 36.73±1.33 | 37.19±5.75 | 26.86±2.34 |
| **#7** | 146.70±53.19 | 64.98±8.46 | 49.64±1.05 | 69.04±3.48 | 62.36±3.40 | 46.87±2.15 |
| **#8** | 43.49±4.97 | 29.89±1.95 | 18.10±1.21 | 25.40±1.64 | 24.98±6.64 | -95.54±6.34[b] |

| Event | δ¹⁸O | δ²H | Ca²⁺ | NO₃⁻ | SO₄²⁻ | EC |
|---|---|---|---|---|---|---|
| #3 | -10.71±0.03 | -73.57±0.18 | 166.28±1.62 | 12.84±0.10 | 23.77±1.47 | 711.20±14.28 |
| #4 | -10.77±0.05 | -73.74±0.32 | 158.11±1.53 | 11.18±0.08 | 21.84±1.35 | 667.00±13.88 |
| #5 | -10.78±0.03 | -74.24±0.20 | 163.97±1.66 | 12.94±0.18 | 24.46±1.51 | 700.40±14.01 |
| #6 | -11.25±0.03 | -77.30±0.21 | 137.90±1.86 | 8.05±0.10 | 14.23±0.92 | 562.20±12.61 |
| #7 | -10.93±0.03 | -75.52±0.18 | 163.61±1.57 | 11.52±0.12 | 21.05±1.30 | 694.40±13.93 |
| #8 | -11.06±0.04 | -76.20±0.21 | 166.67±1.61 | 11.66±0.10 | 22.22±1.37 | 696.80±14.09 |
| **Event** | **Event-water end member ($C_E$) ±$SE_{CE}$ at peak flow** | | | | | |
| #1 | -13.00±0.26 | -91.86±3.65 | 13.14±3.19 | 0.38±0.22 | 0.12±0.06 | 53.88±24.97 |
| #2 | -5.62±0.30 | -55.71±2.57 | 15.90±2.37 | 1.63±0.66 | 1.27±0.63 | 15.80±20.69 |
| #3 | -8.42±0.73 | -61.52±4.44 | 16.31±0.89 | 0.46±0.13 | 0.07±0.04 | 24.02±20.34 |
| #4 | -10.70±0.54 | -94.29±4.36 | 10.05±1.69 | 1.84±0.30 | 0.05±0.04 | 5.91±20.07 |
| #5 | -5.92±0.09 | -46.96±1.32 | 13.76±1.01 | 3.39±0.71 | 2.39±0.60 | 12.87±20.07 |
| #6 | -11.87±0.40 | -86.22±2.21 | 6.99±3.76 | 0.31±0.16 | 0.05±0.04 | 10.58±20.92 |
| #7 | -14.58±1.06 | -130.11±6.16 | 10.81±2.21 | 1.28±0.50 | 0.06±0.04 | 18.15±21.15 |
| #8 | -15.29±0.50 | -121.41±2.76 | 12.15±2.56 | 2.72±0.45 | 0.17±0.15 | 20.24±20.98 |
| **Event** | **Streamwater-end member ($C_S$) ±$SE_{CS}$ at peak flow** | | | | | |
| #1 | -11.85±0.03 | -82.19±0.17 | 78.00±0.78 | 4.16±0.03 | 7.12±0.46 | 414.00±8.28 |
| #2 | -10.27±0.03 | -70.60±0.17 | 64.60±0.66 | 3.73±0.03 | 5.12±0.34 | 264.00±5.28 |
| #3 | -10.04±0.03 | -69.28±0.17 | 89.92±0.89 | 6.26±0.04 | 10.52±0.67 | 346.00±6.92 |
| #4 | -12.54±0.03 | -87.74±0.17 | 90.82±0.90 | 5.39±0.04 | 8.00±0.52 | 428.00±8.56 |
| #5 | -10.20±0.03 | -68.66±0.17 | 98.23±0.96 | 9.94±0.04 | 13.56±0.85 | 542.00±10.84 |
| #6 | -11.58±0.03 | -80.06±0.17 | 107.28±1.04 | 5.48±0.04 | 9.57±0.61 | 336.00±7.24 |
| #7 | -15.12±0.03 | -106.78±0.17 | 91.31±0.90 | 4.58±0.04 | 7.59±0.49 | 366.00±7.32 |
| #8 | -12.54±0.03 | -85.85±0.17 | 142.95±1.36 | 9.65±0.04 | 16.71±1.04 | 1338.00±26.76 |
| | **Event-water fraction $F_E$ ±$SE$ (%) at peak flow** | | | | | |
| **Event** | **δ¹⁸O** | **δ²H** | **Ca²⁺** | **NO₃⁻** | **SO₄²⁻** | **EC** |
| #1 | 42.85±5.80 | 42.29±9.28 | 56.67±1.42 | 65.45±1.35 | 65.21±3.20 | 45.01±2.64 |
| #2 | 6.72±1.49 | 15.68±2.84 | 67.70±1.20 | 82.84±4.50 | 83.17±2.95 | 64.67±2.17 |
| #3 | 29.21±9.43 | 35.61±13.23 | 50.92±0.85 | 53.15±0.75 | 55.93±3.92 | 53.15±2.11 |
| #4 | -[a] | 68.15±14.48 | 45.45±0.98 | 61.99±2.06 | 63.53±3.28 | 36.15±2.16 |
| #5 | 11.79±0.85 | 20.44±1.30 | 43.77±0.94 | 31.43±2.69 | 49.40±5.35 | 23.04±2.32 |
| #6 | 52.06±34.19 | 30.90±8.05 | 23.39±1.51 | 33.24±1.24 | 32.92±6.13 | 36.29±2.40 |
| #7 | 114.79±33.35 | 57.27±6.47 | 47.32±1.05 | 67.77±3.34 | 64.11±3.24 | 48.56±2.15 |
| #8 | 35.03±4.22 | 21.36±1.40 | 15.35±1.27 | 22.44±1.49 | 25.02±6.63 | -94.77±6.38[b] |

[a] Unrealistic event-water fractions were obtained because the δ¹⁸O signatures in precipitation and streamwater were too similar.

[revised manuscript text omitted]